# A propolis-derived small molecule ameliorates metabolic syndrome in obese mice by targeting the CREB/CRTC2 transcriptional complex

Yaqiong Chen[1,8], Jiang Wang [2,8], Yibing Wang[2,3], Pengfei Wang[3,4], Zan Zhou[3,4], Rong Wu[3,4], Qian Xu[5], Hanyun You[6], Yaxin Liu[1], Lei Wang[1], Lingqin Zhou[1], Yuting Wu[4], Lihong Hu[6,9✉], Hong Liu [2,9✉] & Yi Liu [1,7,9✉]

The molecular targets and mechanisms of propolis ameliorating metabolic syndrome are not fully understood. Here, we report that Brazilian green propolis reduces fasting blood glucose levels in obese mice by disrupting the formation of CREB/CRTC2 transcriptional complex, a key regulator of hepatic gluconeogenesis. Using a mammalian two-hybrid system based on CREB-CRTC2, we identify artepillin C (APC) from propolis as an inhibitor of CREB-CRTC2 interaction. Without apparent toxicity, APC protects mice from high fat diet-induced obesity, decreases fasting glucose levels, enhances insulin sensitivity and reduces lipid levels in the serum and liver by suppressing CREB/CRTC2-mediated both gluconeogenic and SREBP transcriptions. To develop more potential drugs from APC, we designed and found a novel compound, **A57** that exhibits higher inhibitory activity on CREB-CRTC2 association and better capability of improving insulin sensitivity in obese animals, as compared with APC. In this work, our results indicate that CREB/CRTC2 is a suitable target for developing anti-metabolic syndrome drugs.

[1] Key Laboratory of Metabolism and Molecular Medicine, the Ministry of Education, Department of Biochemistry and Molecular Biology, School of Basic Medical Sciences, Fudan University, 200032 Shanghai, China. [2] State Key Laboratory of Drug Research and CAS Key Laboratory of Receptor Research, Shanghai Institute of Materia Medica, Chinese Academy of Sciences, 201203 Shanghai, China. [3] University of Chinese Academy of Sciences, No.19A Yuquan Road, 100049 Beijing, China. [4] Key Laboratory of Nutrition and Metabolism, Institute for Nutritional Sciences, Shanghai Institutes for Biological Sciences, 200031 Shanghai, China. [5] Department of Endocrinology, the First Affiliated Hospital of Harbin Medical University, 150081 Heilongjiang Province, China. [6] Jiangsu Key Laboratory for Functional Substance of Chinese Medicine, School of Pharmacy, Nanjing University of Chinese Medicine, 210023 Nanjing, China. [7] State Key Laboratory of Medical Neurobiology and MOE Frontiers Center for Brain Science, School of Basic Medical Sciences, Fudan University, 200032 Shanghai, China. [8]These authors contributed equally: Yaqiong Chen, Jiang Wang. [9]These authors jointly supervised this work: Lihong Hu, Hong Liu, Yi Liu. ✉email: lhhu@njucm.edu.cn; hliu@simm.ac.cn; liuyee@fudan.edu.cn

As a resinous complex mixture, propolis contains natural materials collected by bees from tree buds, sap flows, or other botanical sources and excretions from a bee's palate gland. Having been used as a traditional folk medicine to treat multiple diseases for centuries[1–5], propolis has been found to ameliorate diabetic complications by decreasing glucose and lipid levels[6–8]. Although a few characteristic compounds in propolis, such as polyphenols[9], flavonoids[10,11], and caffeic acid[12,13], have been reported to improve metabolic syndrome, it still remains a serious challenge to determine the new targets of active compounds from propolis and explore the underlying mechanisms.

Obesity-induced insulin resistance is believed to be the major risk factor for developing type 2 diabetes. Patients with this disease usually possess fasting hyperglycemia that is mainly due to the pathologically excess production of glucose from the liver by gluconeogenesis[14,15]. By trigging intercellular cAMP accumulation, glucagon induces CREB phosphorylation at Ser133 via PKA catalytic subunit. In parallel, glucagon also promotes the dephosphorylation of CRTC2 (CREB regulated transcriptional coactivator 2), which leads to its nuclear translocation and subsequent binding with phosphorylated CREB, which allows it to occupy CRE sites in target gene promoters[16]. The CREB/CRTC2 complex increases the transcription of key gluconeogenic genes, such as Pgc-1α, Pck1, and G6pc, to enhance glucose product and output from the liver[17–20]. The hyperactivation of CRTC2 has been shown to be associated with hyperglycemia in both high-fat diet-induced obese (DIO) and db/db mice[19,21]. Conversely, the abolishment of CREB or CRTC2 activity leads to amelioration of insulin resistance[22,23]. In addition, a liver-specific knockout of Crtc2KO has been reported to improve energy metabolism of the whole body through increasing FGF21 protein levels in mice[24]. Moreover, cholesterol synthesis has been demonstrated to be related to high levels of hepatic CRTC2 in humans. Thus, the complex of CREB/CRTC2 has been suggested as a co-regulator of glucose and lipid metabolism[25]. Considering the involvement of cytosol CRTC2 in SREBP1c proteolysis[26], it may not be a practical strategy to improve glucose/lipid metabolism by nonspecific reduction of CRTC2 protein amounts. Taken together, we thought that the disruption of CREB/CRTC2 complex formation would be a more reasonable therapeutic approach to improve glucose/lipid metabolism. In our preliminary study, we identified some natural inhibitors of CREB/CRTC2 in propolis extracts. To characterize these native CREB/CRTC2 inhibitors, we then established a platform to select, identify, and confirm these novel inhibitors targeting CREB/CRTC2 interaction. Using this system, we here report to identify a natural inhibitor, artepillin C, and a developed compound, **A57** that improve insulin sensitivity in obese mice by inhibiting CREB–CRTC2 interaction.

## Results

**Identification of APC as an inhibitor of CREB/CRTC2 protein interaction**. We first verified the anti-diabetic effects of Brazilian green propolis that we purchased on animals with hyperglycemia. As expected, one oral dose of Brazilian green propolis (250 mg/kg) significantly attenuated fasting blood glucose levels in db/db mice (Supplementary Fig. 1a). Then we processed our intravenous glucose (IGTT) and insulin (ITT) tolerance test on db/db mice with oral administration of propolis (250 mg/kg) or vehicle one time daily for 1 week, and pyruvate tolerance test (PTT) on DIO mice with oral administration of propolis (250 mg/kg) or vehicle one time daily for 3 weeks. Expectedly, 1 week of oral administration of propolis remarkably increased insulin sensitivity, as indicated by glucose- and insulin-tolerance tests (IGTT and ITT, Fig. 1a, left and middle). The capacity for hepatic gluconeogenesis, as demonstrated by a pyruvate tolerance test (PTT),

was also significantly decreased in DIO mice with propolis treatment, as compared with those treated with control vehicle (Fig. 1a, right). Using in vivo imaging analysis, we determined that propolis remarkably inhibited the activity of a luciferase reporter driven by the G6pc promoter (G6p-Luc) in the liver of DIO mice (Fig. 1b). Correspondingly, propolis significantly reduced the mRNA levels of gluconeogenic genes including G6pc and Pck1, in the liver of db/db mice (Supplementary Fig. 1b). In line with our in vivo results, propolis markedly decreased CRE-driven luciferase activity in primary hepatocytes exposed to glucagon (Fig. 1c). In addition, propolis exerted little effect on luciferase enzymatic activity by itself (Supplementary Fig. 1d).

Next, we examined the effects of propolis on the cAMP-CREB/CRTC2 signaling axis (Supplementary Fig. 1c) in primary hepatocytes. Unexpectedly, propolis had little effect on the phosphorylation of CREB and the cAMP accumulation induced by glucagon (Fig. 1d). Previous studies have shown that the recruitment of Serine 171-dephosphorylated CRTC2 by CREB to target gene promoters was required for their CRE-driven transcriptional activity[17], which prompted us to examine the effects of propolis on the interaction between CREB and CRTC2. To achieve this, we established a mammalian two-hybrid assay that contained plasmids expressing a constitutively active CRTC2 mutant (CRTC2-S171A)[17] and CREB, fused to the VP16 activation domain (AD) and GAL4-binding domain (BD), respectively, as well as a GAL4-binding-element-driven luciferase reporter (GAL4-Luc), which was only induced by the restored transcriptional activity of the AD being spatially close to BD when CRTC2-S171A bound to CREB (Fig. 1e, top). As expected, propolis remarkably reduced the activity of this GAL4-Luc reporter, compared to a vehicle as agent control (Fig. 1e, bottom), indicating the inhibition of CREB interaction with CRTC2 by propolis treatment.

In order to trace the active compounds in Brazilian green propolis, we first examined the inhibitory activity of propolis fractions isolated by petroleum ether, EtOAc, n-BuOH, or water using a G6p-Luc reporter assay and found that the EtOAc fraction possessed the strongest inhibitory activity (Supplementary Fig. 1e). Next, we screened the 21 compounds purified from this EtOAc fraction (Supplementary Fig. 1f), and found a set of derivatives of cinnamic acid (P1, P2, P3, P5, and P6, Fig. 1g) that significantly inhibited the reporter activity of our two-hybrid assay (Fig. 1f). Among them, P3, named artepillin C (3, 5-diprenyl-4-hydroxycinnamic acid, APC, Fig. 1g), exhibited the most potent capability to dissociate the CREB/CRTC2 complex (Fig. 1f). As artepillin C has been reported to be one of the major and characteristic constituents of Brazilian green propolis (up to 6%)[27,28], APC might play a crucial role in blocking CREB–CRTC2 interaction, which contributes to the inhibition of hepatic gluconeogenesis by propolis.

**APC directly binds to CREB to block its interaction with CRTC2**. To further confirm whether APC disrupts the protein association between CREB and CRTC2, we performed GST-pull-down assays in which GST-CREB beads were used to immuno-precipitated FLAG-tagged CRTC2 proteins that were expressed in HEK293T cells. As expected, APC attenuated the amounts of FLAG-CRTC2 proteins recovered by GST-CREB beads from HEK293T cell lysates (Fig. 2a). Consistently, APC greatly decreased the amount of HA-tagged CRTC2 immunoprecipitated by an antibody recognizing endogenous CREB from the lysates of primary hepatocytes in the presence or absence of glucagon (Fig. 2b). Interestingly, the total immunoprecipitated CRTC2 protein was further reduced by including APC in immunoprecipitation mixtures (Fig. 2b). Moreover, APC treatment significantly decreased the endogenous CREB protein amount co-

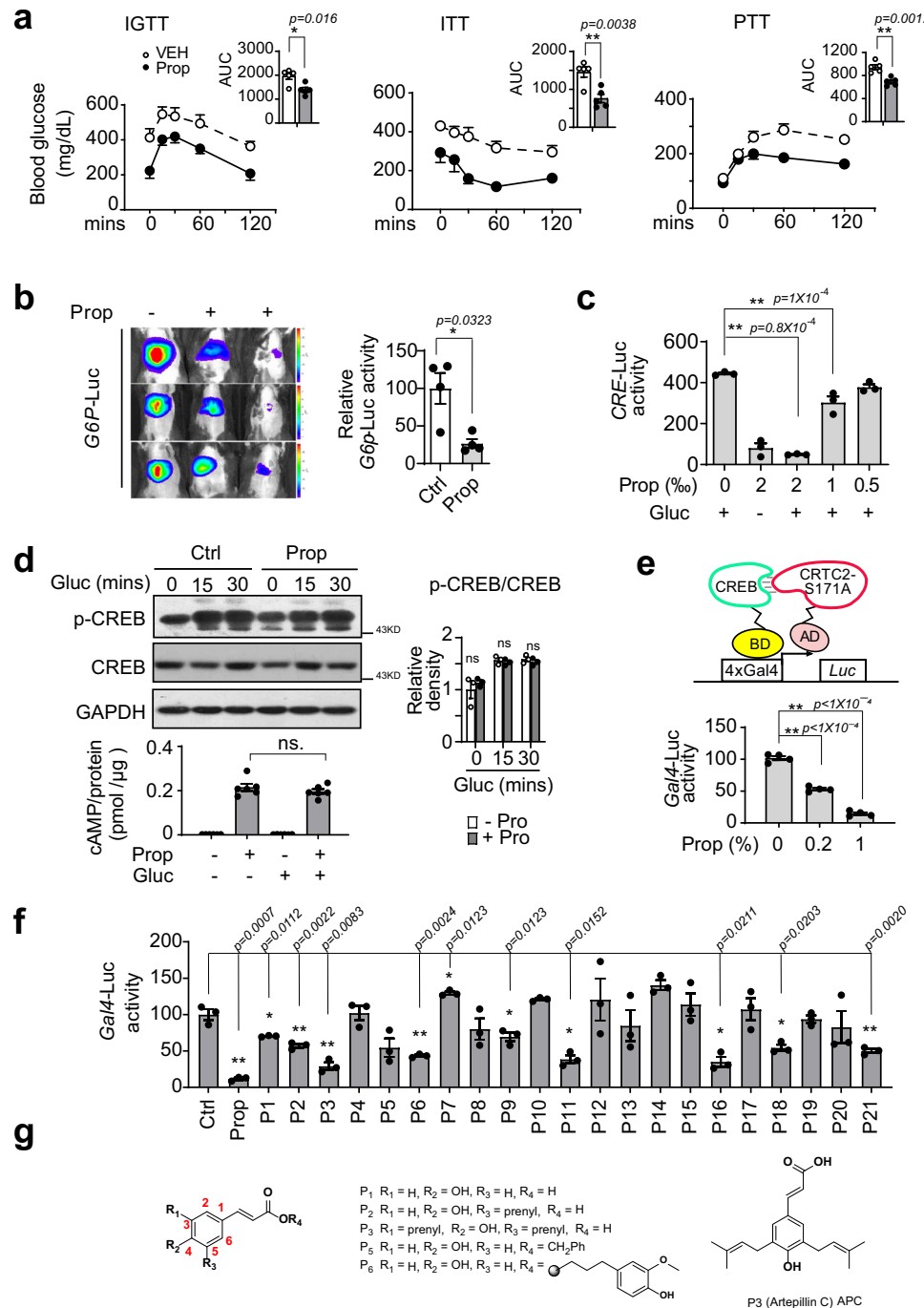

immunoprecipitated by an antibody recognizing endogenous CRTC2 from primary hepatocyte lysates (Fig. 2c). To examine whether APC directly blocked protein interactions between CREB and CRTC2, we set up an in vitro pull-down assay using purified GST-CREB and HIS-CRTC2-S171A protein. In line with the above results, APC significantly diminished the amounts of HIS-CRTC2-S171A recovered by GST-CREB in a dose-dependent manner (Fig. 2d). In contrast, APC had little effect on the association of GST-CREB with HA-P300 (Supplementary Fig. 2a), as well as the interaction between endogenous CRTC2 and CBP proteins (Fig. 2c). Moreover, the interactions of CREB and the isoforms CRTC1 and CRTC3 were also attenuated by APC in overexpression HEK293T cells (Supplementary Fig. 2b, c). Together, these results showed that APC selectively disrupted the formation of CREB with CRTC protein complexes.

To trace the molecular target of APC, we performed surface plasmon resonance (SPR) to investigate the molecular affinity between APC and CREB or CRTC2. As expected, APC directly bound with immobilized HIS-CREB proteins in a dose-dependent manner (Supplementary Fig. 2d). However, no affinity between APC and CRTC2 was detected in this assay (Supplementary Fig. 2e). We further monitored the dynamics of heat release when APC bound to CREB by isothermal titration calorimetry (ITC). When HIS-CREB proteins were titrated into an APC solution, the released heat was fitted to a curve by one-site models. The equilibrium dissociation constant ($K_D$) of molecular interaction between APC and CREB was extrapolated to be about $3.2 \pm 1.32\ \mu M$ (Fig. 3a). In parallel, the $IC_{50}$ of APC for disrupting CREB–CRTC2 interaction was about $24.5 \pm 0.5\ \mu M$, as determined by a dose-response curve from our cell-based two-hybrid

**Fig. 1 APC, an inhibitor of CREB/CRTC2 interaction identified from Brazil propolis. a** Intravenous glucose (IGTT, left) and insulin (ITT, middle) tolerance test of *db/db* mice with oral administration of propolis (250 mg/kg) or vehicle one time daily for 1 week, and pyruvate tolerance test (PTT, right) of DIO mice with oral administration of propolis (250 mg/kg) or vehicle one time daily for 3 weeks. Area under curve analysis (AUC) results for each test are shown as bar graphs respectively. Data are represented as mean ± SEM. (n = 5–7 mice per group); P values were determined by two-way ANOVA followed Bonferroni's multiple comparisons test in curve analysis, or by unpaired two-tailed t test with Welch's correction in AUC analysis. **b** Live imaging assay of *G6p-Luc* reporter activity in DIO mice. C57BL6 mice were induced by high-fat diet for 13 weeks and infected by virus AD-*G6p-Luc* and AD-RSV-ß-*Ga* through tail vein injection, then orally treated with propolis (Prop, 250 mg/kg) or vehicle one time per day for 3 days before live image (left, n = 4 mice per group, 3 shown here). Bar graphs showing *G6P-Luc* activity normalized by β-galactosidase activity (right). Data are represented as mean ± SEM; P values were determined by unpaired two-tailed t test with Welch's correction. **c** The activity of *CRE-LUC* in primary hepatocytes pretreated with propolis (0.05–0.2%) for 1 h prior to 6 h stimulation by glucagon (Gluc, 100 nM). (n = 3 per treatment). Data are represented as mean ± SEM; P values were determined by one-way ANOVA followed Dunnett's multiple comparisons test. One of three independent experiments presented here. **d** Immunoblotting analysis showing phosphorylated CREB (top and right) and measurement of cAMP levels (bottom) in primary hepatocytes pretreated with propolis (0.2%) for 1 h prior to 30-min stimulation by glucagon (100 nM). One representative result from three independent experiments is shown here, and relative P-CREB normalized by CREB is presented as a bar graph (right). Data are represented as mean ± SEM (n = 3); P values were determined by two-way ANOVA followed Bonferroni's multiple comparisons test. **e** The reporter activity of CREB–CRTC2 two-hybrid assay in HEK293T cells incubated with propolis (0–0.1%) for 6 h (bottom, n = 3 per treatment). Top is a schematic diagram of a two-hybrid assay based on CREB/CRTC2 interaction. One representative result from three independent experiments is shown. Data are represented as mean ± SEM; P values were determined by one-way ANOVA followed Dunnett's multiple comparisons test. **f** The reporter activity of a CREB–CRTC2 two-hybrid assay in HEK293T cells treated with propolis (0.2%), vehicle (DMSO), or isolated compound (25 μM) as indicated (n = 3 per treatment). One representative result from two independent experiments is shown. Data are represented as mean ± SEM; P values were determined by one-way Brown–Forsyth and Welch ANOVA followed Dunnett's multiple comparisons test. **g** Molecular architectures of P3 and its analogues, P1, P2, P3, P5, and P6. ns, P > 0.05; *P < 0.05; **P < 0.01. Source data for this figure are provided as a Source data file.

assay (Fig. 3b). Meanwhile, no effect of APC on luciferase enzymatic activity was detected (Supplementary Fig. 2f). In addition, an analogs compound, P4, (*p*-coumaric acid ethyl ester), isolated from propolis, neither directly bound with HIS-CREB proteins nor decreased CREB/CRTC2 interaction (Supplementary Fig. 2g and Fig. 1f), which excluded a nonspecific interaction between propolis small molecular compounds with CREB protein. Taken together, the above evidence demonstrated that CREB was a direct target of APC.

We further explored the mechanism of APC inhibition of CREB-driven transcription. We found that APC inhibited FLAG-CRTC2 binding with the MYC-CREB-ZIP domain, which was the affinity region of CREB recognized by CRTC2 (Fig. 3c). As P300/CBP is another coactivator of CREB, we wondered whether APC disrupted the interaction between CREB and CBP/P300. Our results suggested that APC had little effect on the interaction between either full-length P300 and CREB proteins as demonstrated by co-immunoprecipitation (Supplementary Fig. 2a), or CBP-KIX and CREB-KID domains as determined by two-hybrid assay (Supplementary Fig. 2h). However, APC dose-dependently reduced the acetylation level of H3K27ac, which are induced by CBP (Fig. 3d). These results suggested that APC inhibited the transcriptional activity of the CREB–CRTC2-CBP complex by reducing CRTC2 interaction and decreasing the acetyltransferase activity of CBP.

**APC attenuates gluconeogenesis in primary hepatocytes.** Considering the pivotal role of CREB/CRTC2 in regulating hepatic gluconeogenesis, we next examined whether APC was capable of modulating glucose production in vitro. In line with the inhibitory effect of APC on CREB–CRTC2 interaction, the amount of CRTC2 proteins recruited to the cAMP response element (CRE) sites on the promoters of *G6pc*, *Pgc-1α*, and *Pck1* were significantly attenuated by APC (10 μM) in primary hepatocytes exposed to glucagon (Fig. 4a). In contrast, APC exerted little effect on the occupancy of CREB or CBP at CRE sites under the same conditions (Fig. 4a and Supplementary Fig. 3a). Consistently, APC remarkably decreased the mRNA accumulation of *G6pc*, *Pck1*, and *Pgc-1α* induced by glucagon in primary hepatocytes (Fig. 4b). As a result, APC (5 μM) reduced glucose output from primary hepatocytes induced by glucagon (Fig. 4c). In addition, such an inhibitory effect was not due to cell toxicity, as

APC (up to 100 μM) did not impair hepatocyte viability (Supplementary Fig. 3b).

We then explored the underlying mechanism whereby APC reduces the gluconeogenic program. Consistent with the effects of propolis, APC did not alter the cAMP accumulation stimulated by glucagon (Supplementary Fig. 3c). In line with this, APC had very little effect on CREB phosphorylation, CRTC2 dephosphorylation, or nuclear translocation (Fig. 4d and Supplementary Fig. 3d), which excluded these important events of the glucagon-CREB/CRTC2 signaling pathway from being targets of APC. Moreover, these observed effects of APC do not cross-talk with the AMPK signaling pathway, because this compound (up to 200 μM) did not induce phosphorylation of AMPK in primary hepatocytes (Supplementary Fig. 3e).

As the S171A mutation has been shown to mimic constitutive activation of CRTC2, we checked the effect of APC on the transcriptional activity of CRTC2-S171A. Expectedly, APC also reduced CRTC2-S171A inducing *G6p*-luciferase activity, as well as the *G6p* and *Pck1* gene mRNA levels in primary hepatocytes (Fig. 4e, f). Conversely, APC was unable to further reduce both glucose output and key gluconeogenic gene (*Pgc-1α* and *Pck1*) expression in CRTC2-null primary hepatocytes exposed to glucagon (Fig. 4g, h), implying the dependency of CRTC2-mediated transcription on the inhibitory effects of APC on glucose metabolism.

**APC ameliorates hyperlipidemia in obese mice.** Having seen the ability of APC to reduce the gluconeogenic profile in primary hepatocytes, we wondered whether this compound could provide metabolic benefits in vivo. Interestingly, we found that intraperitoneal injection (i.p.) of synthesized APC (10 or 20 mg/kg) reduced the body weights of both DIO and *db/db* mice (Fig. 5a and Supplementary Fig. 4a, b), which was mainly due to a significant decrease in fat mass and the fat to lean ratio (Fig. 5b and Supplementary Fig. 4c). Consistently, the levels of triglycerides (TGs) in the serum and liver, total cholesterol (TC), LDL cholesterol (LDL-C), HDL cholesterol (HDL-C), and VLDL cholesterol (VLDL-C) in the serum were reduced in DIO mice treated with APC as compared to those with treated with vehicle alone (Fig. 5c–f). Similar results were also observed in experiments using *db/db* mice (Supplementary Fig. 4c–f). In addition, APC significantly enhanced nonesterified fatty acid (NEFA) levels in

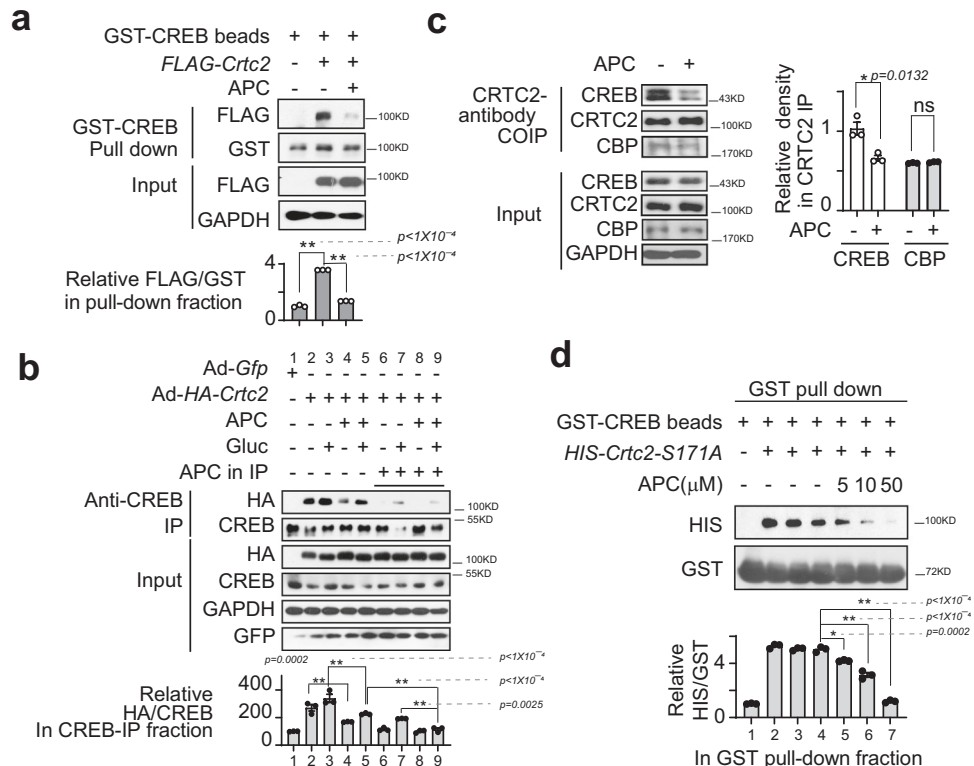

**Fig. 2 APC blocks interaction between CREB and CRTC2. a** Analysis of GST-CREB immunoprecipitated with overexpressed FLAG-CRTC2 from the lysates of HEK293T cells. APC (10 μM) was added to the culture media for 1 h and the following IP mixture. The relative gray density of FLAG in GST-pull-down fraction is shown as a bar graph (bottom). One of three independent experiments is presented (top, $n = 3$). Data are represented as mean ± SEM); $P$ values were determined by one-way ANOVA followed Dunnett's multiple comparisons test. **b** Analysis of HA-CRTC2 immunoprecipitated with endogenous CREB from primary hepatocyte lysates. APC (10 μM) was added to the culture media for 1-h before 30-min of stimulation with glucagon (Gluc, 100 nM) and the following IP mixture as indicated. Top panel presents one of three independent experiments, and the relative gray density of HA in CREB-IP-faction is shown as a bar graph (bottom, $n = 3$). Data are represented as mean ± SEM; $P$ values were determined by one-way ANOVA followed Dunnett's multiple comparisons test. **c** Immunoblotting analysis of endogenous CREB, CRTC2, and CBP interaction by an anti-CRTC2 antibody from the lysate of primary hepatocytes. APC (10 μM) was added to the culture media for 1-h before 30-min stimulation with glucagon. Related CREB and CBP normalized by CRTC2, respectively, in CRTC2-IP fractions. The top panel presents one of three independent experiments, and the relative gray density of CREB and CBP normalized to CRTC2 in the CRTC2-IP-faction is shown as a bar graph (bottom, $n = 3$). Data are represented as mean ± SEM; $P$ values were determined by two-way ANOVA followed Bonferroni's multiple comparisons test. **d** Immunoblotting analysis of protein GST-CREB recovered HIS-CRTC2-S171A proteins purified from *E. coli*. The addition of APC (5, 10, or 50 μM) in pull-down mixtures is indicated. The top part presents one of three independent experiments, and the relative gray density of HIS normalized to GST in GST-pull-down faction is shown as a bar graph (bottom, $n = 3$). Data are represented as mean ± SEM; $P$ values were determined by one-way ANOVA followed Dunnett's multiple comparisons test (compared with APC = 0).

the plasma (Fig. 5g), and APC also reduced NEFA (Fig. 5h) and glycerol levels (Fig. 5i) in the liver of DIO mice. Consistent with these data, histological data showed that APC remarkably reduced the sizes of lipid droplets in the liver and adipose tissues of DIO and *db/db* mice (Fig. 5j and Supplementary Fig. 4g).

As CREB and CRTC2 are constitutively expressed in various tissues, we wondered whether the liver was the main target organ of APC in vivo. In fact, 2–8 h after treatment, the content of APC in the liver was the highest as compared with other indicated tissues, including the kidney, muscle, brain, spleen, pancreas, white and brown fat. Moreover, the peak concentration of APC in the liver was about 916.8 ± 207.3 ng/g (3.06 ± 0.69 μM) after 4-h of exposure to this compound (Fig. 5k), which was comparable to the $K_D$ between APC and CREB as determined in vitro (Fig. 3a). Although the relative CRTC2 protein level in the liver was not the highest of the tissues/organs we checked (Supplementary Fig 4h), the normalized level of APC content relative to the CRTC2 protein level in each tissue/organ was the highest in the liver among these tissues/organs (Fig. 5l). Moreover, we checked the transcriptional levels of CREB/CRTC2-specifically regulated genes (e.g., *Nr4a1*, *Pdk1*, *Gcg*, and *C2cd4a*) in various tissues[29–32],

including the muscle, brain, spleen, kidney, small intestine, and pancreas, of C57 wild-type (WT) mice that were orally administered one dose of APC (20 mg/kg) or vehicle control. As shown in Supplementary Fig. 4i, APC did not significantly affect the mRNA levels of CREB/CRTC2 regulated genes in these tissues, which is consistent with the limited APC distribution in these tissues.

**APC reduced SREBP-mediated lipid synthesis**. The liver is frequently challenged by fluctuations of lipid overdose and demand, which highlights SREBP-mediated hepatic lipid metabolism for the maintenance of whole-body lipid homeostasis[33–36]. As master regulators of lipid homeostasis, SREBP-1a and SREBP-1c are involved in the synthesis of lipid and fatty acids, respectively, and SREBP-2 takes part in cholesterol metabolism[34,37,38]. Having seen the reduction of TG and cholesterol levels in the serum and liver by APC treatment, we wondered whether APC modulated the hepatic SREBPs signaling pathway. Indeed, APC significantly decreased the mRNA levels of hepatic *Srebp-1a*, *1c*, and *2* (Fig. 6a), as well as reducing the accumulations of their mature proteins in the livers of DIO mice (Fig. 6b). Moreover, APC deceased the

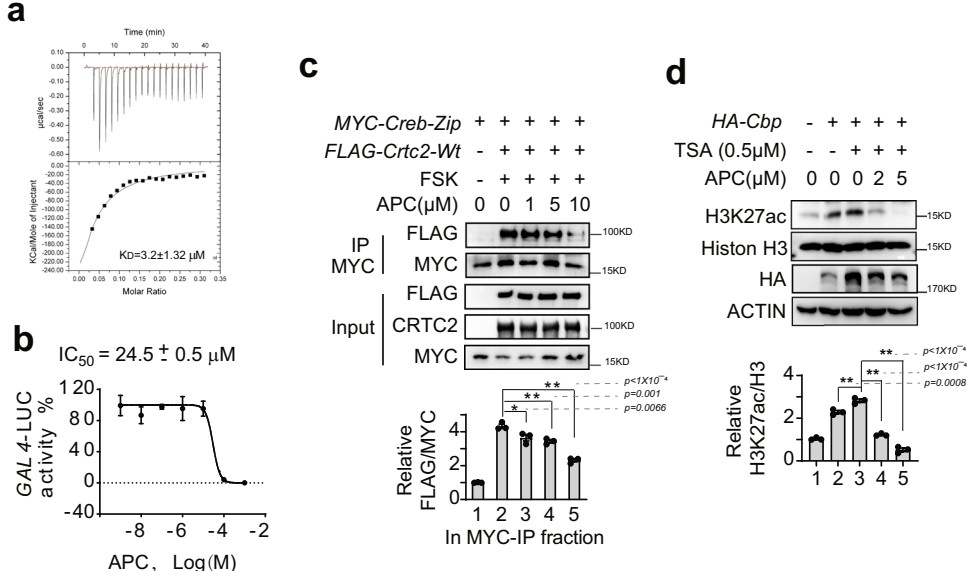

**Fig. 3 APC inhibits CRTC2 interacting with CREB-ZIP and reduces acetyltransferase activity of CBP by binding with CREB protein directly. a** Isothermal titration calorimetry (ITC) of HIS-CREB protein solution titrated into APC solution. The bottom solid line represents a nonlinear least-squares fitted by a single-site binding model. One of two independent experiments is shown here. **b** $IC_{50}$ of APC inhibitory activity detected by CREB–CRTC2 two-hybrid assay in HEK293T cells ($n = 3$ per treatment, data are represented as mean ± SEM). One of three independent experiments is shown here. **c** Immunoprecipitation analysis of FLAG-CRTC2-WT recovered by anti-MYC antibody from HEK293 cells co-transfected with plasmids for a *Myc-Creb-Zip* fragment and the *FLAG-Crtc2-Wt*, stimulated by FSK (10 nM) for 2-h before harvest. APC (1–10 μM) was included in immunoprecipitation mixtures as indicated. Top represents one of three independent experiments, and the relative gray density of FLAG normalized to MYC in MYC-IP- factions is shown as a bar graph (bottom, $n = 3$). Data are represented as mean ± SEM; $P$ values were determined by one-way ANOVA followed Dunnett's multiple comparisons test. **d** Immunoblotting analysis of the acetyltransferase activity of CBP. The protein levels of H3K27ac and total Histone H3 were determined using specific antibodies from HEK293 cell lysate overexpressing *HA-Cbp* plasmid. Pretreatment with trichostatin (TSA, 0.5 μM) for 1 h prior to incubation with APC (2, 10 μM) for another 2-h. The top represents one of three independent experiments, and the relative gray density of H3K27ac normalized to total Histone H3 in cell lysate is shown as a bar graph (bottom, $n = 3$). Data are represented as mean ± SEM; $P$ values were determined by one-way ANOVA followed Dunnett's multiple comparisons test. ns, $P > 0.05$; *$P < 0.05$; **$P < 0.01$. Sources data for this figure are provided as source data file.

hepatic mRNA levels of *Scap* but increased these levels of *Insig1* and *2b*, which both suppress the translocation and cleavage of pre-matured SREBPs (Fig. 6c). Accordingly, the hepatic mRNA levels of SREBP target genes, including those involved in cholesterol synthesis (*Hmgcl* and *Hmgcs*), cholesterol uptake (*Ldlr* and *Pcsk9*), and fatty acid synthesis (*Acc, Fasn, Acl*, and *Acsl1*) were reduced by APC treatment in DIO mice (Fig. 6c). In addition, APC increased *Cpt1* and *Scd-1* transcription (fatty acid oxidation), but attenuated *ApoE, ApoB*, and *Mttp* (lipid secretion, Fig. 6c) transcription. Consistent with the reduction of hepatic levels of NEFA and glycerol (Fig. 5g, h), APC increased the mRNA levels of key enzymes involved in TG hydrolysis in the liver, including *Mgll, Cel*, and *Lipg* (Fig. 6c).

Consist with its effects on whole-body lipid homeostasis, APC profoundly affected the transcriptional profile of lipid metabolism in the white adipose tissue (WAT) of DIO mice, including attenuated expression of *Srebp-1c, 2* (transcription factors), *Fasn, Acc, Scd-1* and *Acsl1* (fatty acid synthesis), *ApoB* (fatty acid-binding protein), *Lpl* (lipogenesis), *Retn* (WAT secretory factor, resistin). APC also increased the expression of *Cpt1* (fatty acid oxidation), *Hsl, Atgl1* (lipolysis), and *Glut4* (glucose absorption, Fig. 5d). Whereas, the mRNA levels of master transcriptional factors (*Ppary*) and regulators of adipogenesis (*Creb, Cebpa*, and *Tnfaip2*) were not affected by APC treatment (Fig. 6d). We also checked the effects of APC on brown fat tissue (BAT) in DIO mice. As shown in Fig. 6e, APC significantly decreased the mRNA levels of *Srebp-1c, Scd-1*, and *Ucp1*, but increased those of *Cpt1* (fatty acid oxidation), *Hsl, Atgl1, Mgl* (lipolysis), *Glut4* (glucose uptake), as well as *Ucp2* and *Ucp3* (thermogenesis), but had very

little effect on *Ppary* and *Creb*. Thus, in line with the attenuation of TG levels but enhancement of NEFA levels in the plasma and the reduction of lipid droplet size in WAT (Fig. 5c–j), APC decreased the expression of genes in lipid synthesis, but enhanced those involved in lipolysis and fatty acid consumption in the main lipid metabolic organs/tissues.

A previous study reported that CRTC2 interacts with SEC31A, a subunit of COPII, attenuates COPII-dependent pre-SREBP1c vesicle trafficking from the ER to the Golgi, and thus, inhibits SREBP1c-mediated lipogenesis[26]. In this mechanism, CRTC2 interaction with SEC31A in the cytosol was a critical step for the CRTC2- mediated inhibition effect. We wondered whether APC affected the affinity between CRTC2 and SEC31A. Co-immunoprecipitation indicated that APC had no impact on the protein interactions between CRTC2 and SEC31A, which excluded the possibility that APC regulated CRTC2-dependent-pre-SREBP1c trafficking and maturation (Supplementary Fig. 5b). These data also suggested that APC regulated transcriptional function of nuclear CRTC2 but not its cytosol function. Consistently, APC did not further decrease the mRNA levels of *Srebp-1c* and *2* in livers collected from CRTC2 knockout mice (Fig. 6f). In addition, we did not detect APC further reducing the protein levels of SREBP1 and 2 in the liver of *Crtc2*-KO mice (Fig. 6g and Supplementary Fig. 5a). Correspondingly, APC also had little effect on plasma and liver cholesterol as well as on liver TG levels in *Crtc2*-KO mice (Fig. 6h and Supplementary Fig. 5c, d). Taken together, these data suggest that the capability of APC to decrease the expression of SREBPs required the transcriptional activity of CREB/CRTC2.

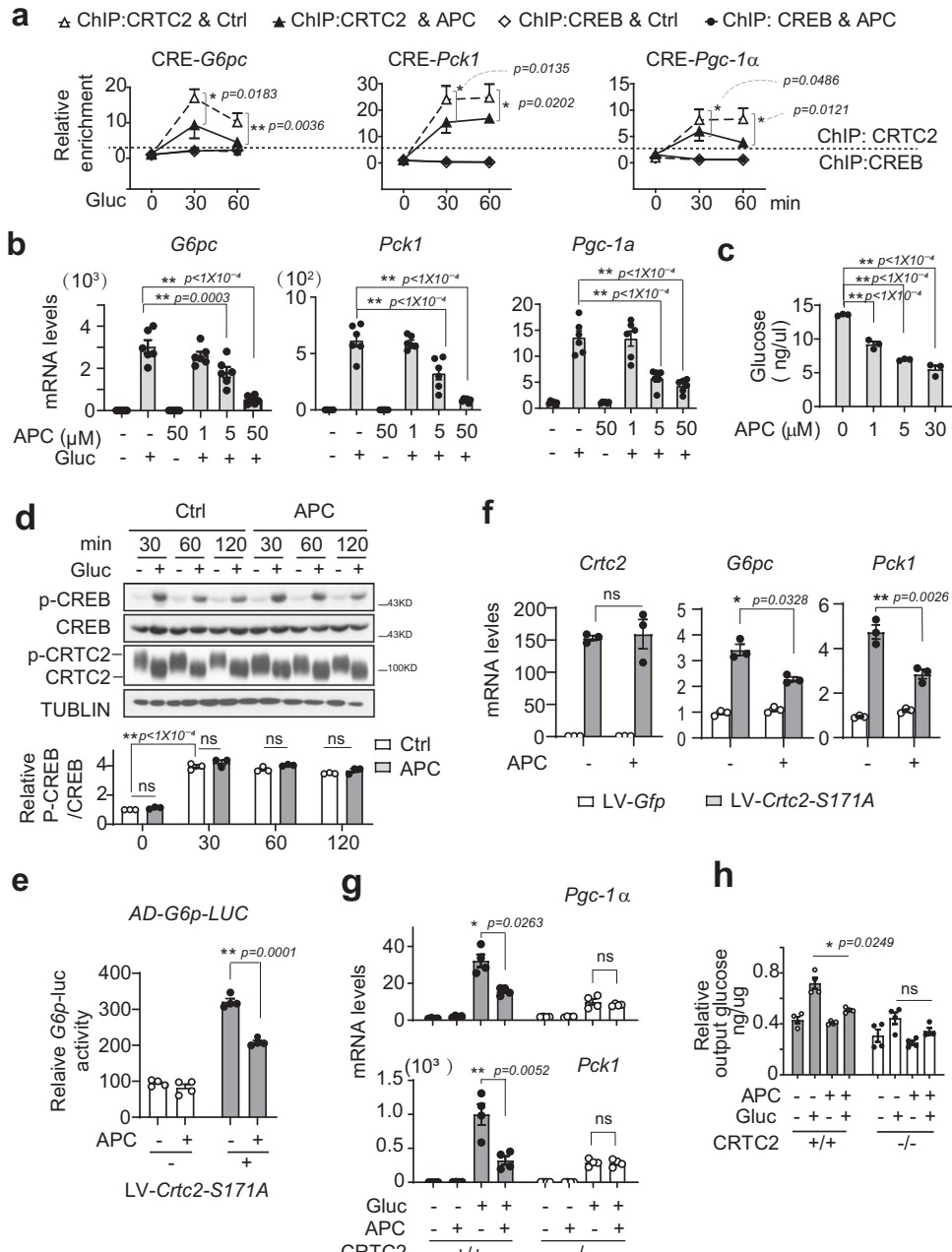

## APC reduces SREBP1 transcription via the CRE-LXRα axis.

CRE sites play a critical role in the transcription of SREBP2[25]; thus, we explored the underlying mechanism for APC decreasing SREBP1 (code by gene *Srebf1*) transcription. Besides positive feedback regulation by a gene's protein product[39,40], the transcription of *Srebf1* has been reported to be also controlled by some nuclear receptors, including LXRα, RXRα, and PPARα[41–43], which form heterodimers to bind an LXRE element in the promoter of *Srebf1*[42,44,45]. In addition, two CREB/CRTC2 targets[14,46], the peroxisome proliferator-activated receptor gamma coactivator-1α, β (PGC-1α, β), have been demonstrated to be transcriptional coactivators of these nuclear receptors and active SREBP1 is recruited at SREs[47–49] (top panel in Fig. 7c). We then checked whether the transcription of these nuclear receptors/coactivators was altered by APC. qPCR analysis showed that *Pgc-1α* and *Pgc-1β* expression was significantly reduced by APC in the liver, white and brown adipose tissues from DIO mice (Fig. 6c–e), which suggested that PGC-1α, β mediated, at least

partially, the inhibitory effect of APC on SREBP1 transcription. Furthermore, APC decreased the mRNA levels of *Lxra*, but had little effect on those of *Pparα*, *Pparγ*, and *Rxrα* in the liver (Fig. 7a). Consistently, APC significantly reduced LXRα protein levels after insulin treatment for 18-h in primary hepatocytes (Supplementary Fig. 6a, b). In addition, APC decreased the protein levels of LXRα and SREBP1 in the liver of DIO mice (Fig. 7b). Correspondingly, ChIP revealed that APC reduced the amounts of LXRα and RXRα recruited to an LXRE element of *Srebf1* promoter in primary hepatocytes (Fig. 7c). These data demonstrated that the *Lxrα* mediated APC inhibition of *Srebf1* transcription.

The above data prompted us to consider whether LXRα was regulated by CREB/CRTC2. Interestingly, we found a conserved half CRE site (−984) in the *Lxra* promoter (Fig. 7d), which suggested that *Lxrα* transcription was probably modulated by CREB/CRTC2 directly. Indeed, ChIP analysis revealed that CREB was present on the half CRE site of the *Lxra* promoter, which was insensitive to either glucagon or APC treatment. On the one

**Fig. 4 Gluconeogenic transcription inhibited by APC. a** ChIP assay of CRTC2 and CREB recruited to the CRE sites in the promoters of *G6pc*, *Pck1*, and *Pgc-1α* in primary mouse hepatocytes overexpressing HA-CRTC2. Pretreatment with APC (10 μM) for 1-h prior to stimulation by glucagon (Gluc, 100 nM) for indicated times ($n = 4$ per treatment). One of three independent experiments is presented here. *$P < 0.05$; **$P < 0.01$; $P$ values were determined by two-way ANOVA followed Bonferroni's multiple comparisons test. **b** Quantitative PCR analysis of *G6pc*, *Pck1*, and *Pgc-1α* gene expression in primary hepatocytes treated with vehicle or APC (1, 5, or 50 μM) for 1-h prior to 4-h stimulation with glucagon (100 nM, $n = 3$ per treatment). One of three independent experiments is presented here. Data are represented as mean ± SEM. *$P < 0.05$; **$P < 0.01$; $P$ values were determined by one-way ANOVA followed Dunnett's multiple comparisons test. **c** Glucose output from primary hepatocytes pretreated with APC (1, 5, or 30 μM) for 1-h prior to 8-h stimulation by glucagon (100 nM, $n = 3$ per treatment). One of three independent experiments is presented here. Data are represented as mean ± SEM. *$P < 0.05$; **$P < 0.01$; $P$ values were determined by one-way ANOVA followed Dunnett's multiple comparisons test. **d** Immunoblotting analysis of phosphorylated CREB and dephosphorylated CRTC2 protein in primary hepatocytes exposed to APC (10 μM) for 1-h before stimulation by glucagon (100 nM) for indicated times (30, 60, or 120 min). One of three independent experiments (top) is presented here and the relative density of P-CREB is normalized to CREB and is shown as a bar graph (bottom, $n = 3$ per treatment). One of three independent experiments is presented here. Data are represented as mean ± SEM. ns, $P > 0.05$; *$P < 0.05$; **$P < 0.01$; $P$ values were determined by two-way ANOVA followed Bonferroni's multiple comparisons test. **e** Luciferase *G6p-Luc* activity. Primary hepatocytes were infected by lentivirus LV-*Crtc2-S171A* as well as AD-*G6p-luc* and AD-RSV-*β-Gal* for 24-h, then incubated with APC (10 μM) for 8-h before luciferase assay ($n = 4$ per treatment). One of three independent experiments is presented here. Data are represented as mean ± SEM. *$P < 0.05$; **$P < 0.01$; $P$ values were determined by two-way ANOVA followed Bonferroni's multiple comparisons test. **f** Quantitative PCR analysis of *Crtc2*, *Pgc-1α*, and *Pck1* gene expression in primary hepatocytes. Infected by lentivirus, *Crtc2-S171A* or *Gfp* were expressed in primary hepatocytes isolated from wild-type C57 mice for 72-h, followed by incubation with APC (10 μM) for 8-h before RNA isolation ($n = 3$ per treatment). One of three independent experiments is presented here. Data are represented as mean ± SEM. *$P < 0.05$; **$P < 0.01$; $P$ values were determined by two-way ANOVA followed Bonferroni's multiple comparisons test. **g** Quantitative PCR analysis of *Pgc-1α* and *Pck1* gene expression in primary wild-type or CRTC2-null hepatocytes exposed to APC (10 μM) for 1-h prior to 4-h stimulation with glucagon (100 nM, $n = 4$ per treatment). One of three independent experiments is presented here. Data are represented as mean ± SEM. *$P < 0.05$; **$P < 0.01$; $P$ values were determined by two-way ANOVA followed Bonferroni's multiple comparisons test. **h** Glucose output from wild-type or CRTC2-null primary hepatocytes exposed to APC (10 μM) for 1-h prior to 6-h stimulation by glucagon (100 nM). One of three independent experiments is presented here. Relative output glucose level normalized by protein content of assay media. Data are represented as mean ± SEM ($n = 4$ per treatment). ns, $P > 0.05$; *$P < 0.05$; **$P < 0.01$; $P$ values were determined by two-way ANOVA followed Bonferroni's multiple comparisons test. Source data for this figure are provided as source data file.

hand, the occupancy of CRTC2 on the same promoter region was significantly induced by glucagon, while being attenuated by APC on the other hand (Fig. 7d). In addition, when the presence of this half CRE site at position −984 was deleted, the inhibition of *Lxrα*-luc reporter activity by APC was attenuated immediately (Fig. 7e). To mimic constant activation of *CRE*-driven transcription, *Crtc2-S171A* was overexpressed in primary hepatocytes by lentivirus. Expectedly, overexpressed *Crtc2-S171A* raised the mRNA levels of *Lxrα* as well as those of *Srebf1* and *Srebf2* (Fig. 7f). Correspondingly, APC effectively reduced the mRNA levels of these genes in the same conditions (Fig. 7f). Moreover, *Crtc2*KO reduced the mRNA and protein levels of *Lxrα* (Fig. 7g, h), as well as decreased the protein levels of SREBP1 and 2 in the liver, (Fig. 6g). Interestingly, APC was not capable of further inhibiting these genes' expression in the livers of *Crtc2*KO mice (Figs. 6g, h and 7g, h). These results suggested that APC inhibition of *Lxrα* transcription was dependent on *CRE*-driving.

In addition, APC did not alter the activity of *PPRE-luc* (PPAR response element-driven luciferase reporter) induced by the overexpression of PPARα, RXRα, or LXRα (Supplementary Fig. 6c), which excluded the agonism/antagonism of these nuclear receptors by APC. Taken together, these data suggested that APC decreased SREBP1 transcription via the CREB/CRTC2-CRE-LXRα−LXRE axis. Associated with the evidence that a CRE site was located in the *Srebf2* promoter[25], we suggest that CREB/CRTC2-mediated APC inhibits the transcription of the genes *Srebf1* and *Srebf2* (Supplementary Fig. 6d).

**APC improves insulin resistance in diabetic mice.** Having seen APC inhibiting gluconeogenesis and lipid synthesis in vitro and in vivo, we then examined the effects of APC on insulin sensitivity in diabetic mice. Indeed, APC substantially decreased the *G6P-Luc* reporter activity in the livers of DIO mice (Fig. 8a). Consistently, APC remarkably reduced the hepatic mRNA levels of *G6pc*, *Pck1*, and *Pgc-1α* in DIO mice and *db/db* mice (Fig. 8b and Supplementary Fig. 7a). Correspondingly, the hepatic protein

levels of PCK1 and PGC-1α were decreased by APC in both DIO mice and *db/db* mice (Fig. 8c and Supplementary Fig. 7b).

Next, we examined the effects of APC on glucose metabolism in these mice. In agreement with the effects on the gluconeogenic process, APC remarkably decreased fasting blood glucose levels (Fig. 8d and Supplementary Fig. 7c) without a change in plasma insulin and glucagon levels in either DIO mice or *db/db* mice (Fig. 8e and Supplementary Fig. 7d–f). However, APC enhanced the levels of phosphorylated AKT protein in the liver, muscle, brown and white adipose tissues in these animals (Supplementary Fig. 7g), demonstrating the amelioration of insulin signaling by APC. Correspondingly, APC greatly enhanced whole-body insulin sensitivity in these mice, as shown by GTT and ITT assays (Fig. 8f, g Supplementary Fig. 7j, k). In addition, the capability for gluconeogenesis, as suggested by PTT assay, was significantly inhibited by APC in these diabetic animals (Fig. 8h and Supplementary Fig. 7l).

Having seen APC decreasing body weight and enhancing insulin sensitivity in vivo, we next investigated whether APC impacted whole-body energy metabolism. Although the physical activities between the two groups were similar, the respiratory exchange ratio (RER) and energy expenditure of *db/db* mice receiving APC were significantly elevated (Fig. 8i), which may have contributed to their decreased body-weight gain. Moreover, no significant effects of APC administration on daily food intake (Fig. 8j and Supplementary Fig. 7h) or the appetite-stimulating hormone ghrelin (Fig. 8k) were observed in these animals. Although the effect of APC on leptin levels in DIO mice was not significant, APC effectively reduced the leptin levels in *db/db* mice (Fig. 8l and Supplementary Fig. 7i). Taken together, our results demonstrated that APC protects mice from the development of insulin resistance by both diet-induction and genetic modification. In contrast to its profound effects on obese mice, APC had little effect on RER, body weight, and NEFA levels in the plasma of lean mice (Supplementary Fig. 8a–c), suggesting that APC only ameliorates excessively increased CREB/CRTC2 activity in obese mice, but does not affect its normal function in lean animals. In

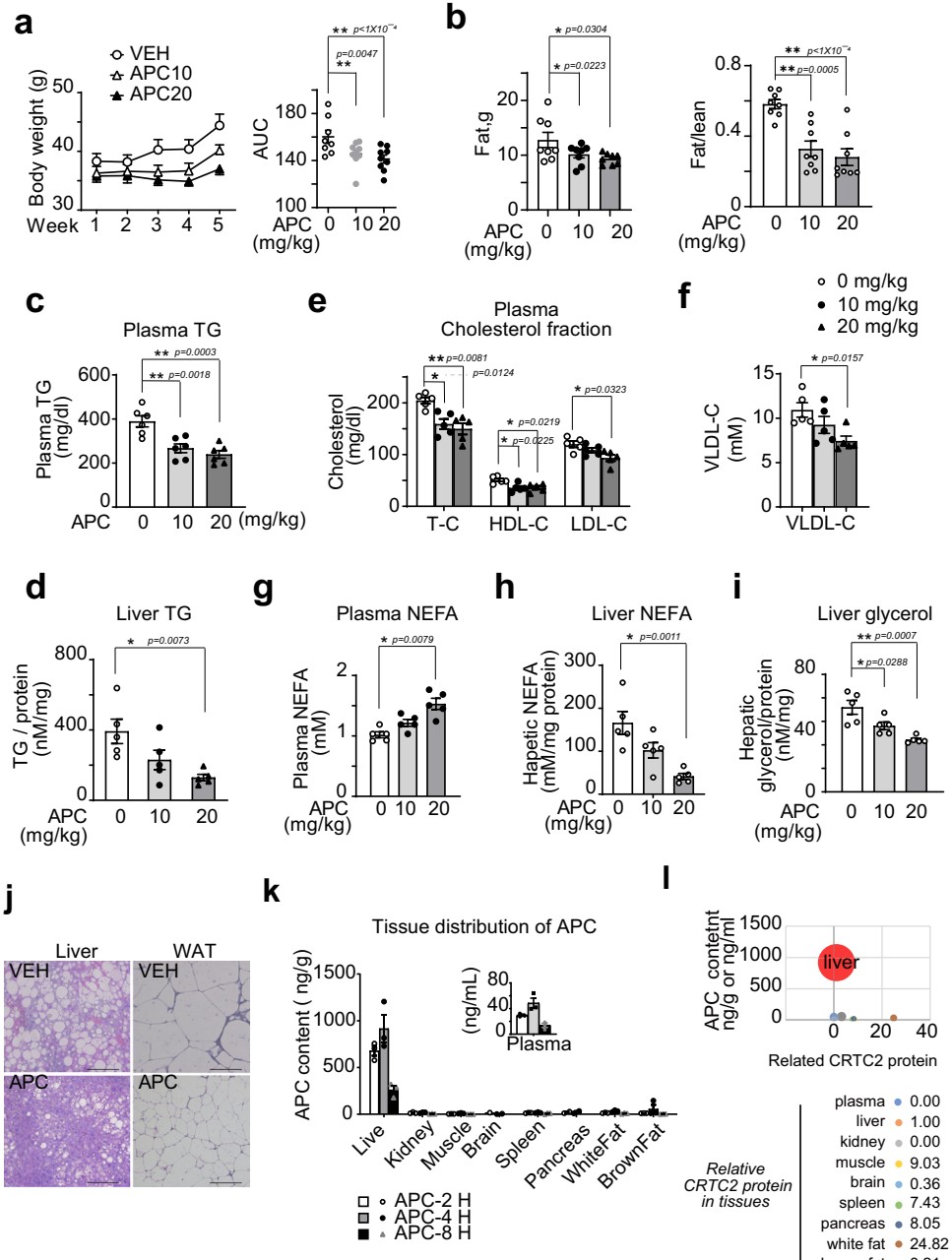

**Fig. 5 APC improves hyperlipidemia of obese mice.** DIO mice were continuously i.p. injected with control vehicle or APC with indicated doses (APC, 0, 10, and 20 mg/kg) or vehicle (Ctrl) one time daily for 5 weeks ($n = 5$–9). **a** Body-weight curves of these mice (left). Area under curve (AUC) analysis of curves is shown as a scatter chart at the right. Data are represented as mean ± SEM ($n = 8$–9 per group). *$P < 0.05$; **$P < 0.01$; $P$ values were determined by two-way ANOVA followed Bonferroni's multiple comparisons test in curve analysis, or by using one-way ANOVA followed Dunnett's multiple comparisons test in AUC analysis. **b** In vivo NMR analysis of whole-body fat mass (left) and fat composition (right, fat/lean ratio) of these mice ($n = 8$ per group). Data are represented as mean ± SEM. *$P < 0.05$; **$P < 0.01$; $P$ values were determined by one-way ANOVA followed Dunnett's multiple comparisons test. The triglyceride (TG) contents in the serum (**c**) and liver (**d**) of these mice. The contents of (**e**) total cholesterol (TC), LDL cholesterol (LDL-C), and HDL cholesterol (HDL-C), (**f**) VLDL-C, and (**g**) nonesterified fatty acid (NEFA) in the serum of these mice. **h** The contents of hepatic NEFA and (**i**) glycerol in these mice, which were normalized by hepatic protein levels as determined by BCA assay. Data are represented as mean ± SEM ($n = 5$–6). *$P < 0.05$; **$P < 0.01$; $P$ values were determined by one-way ANOVA followed Dunnett's multiple comparisons test. **j** Histological analysis of liver and epididymis fat (WAT) from DIO mice treated with APC (20 mg/kg) or vehicle for 5 weeks. Tissues were stained with hematoxylin and eosin. Scale bars represent 100 μm. Here is shown one present result from three independent experiments. **k** The tissue distribution of APC in vivo ($n = 3$ per group, data are represented as mean ± SEM). Two to 8-h after one oral dose (20 mg/kg), the APC content in indicated tissues of C57BL6 was detected by a liquid chromatography-mass spectrometry/mass spectrometry (LC–MS/MS) and normalized by protein concentration. **l** Combination analysis of APC concentration and relative CRTC2 protein level in tissues. Taking the APC concentration (4-h after p.o.) as the $Y$ axis, the relative content of CRTC2 as the $X$ axis, and the APC content as the size of bubbles, each tissue was projected on the graph (top). The relative density of CRTC2 protein of indicated tissues was corrected to TUBULIN and normalized with that in the liver (as base) (bottom, and associated with Supplementary Fig. 4h). Source data for this figure are provided as source data file.

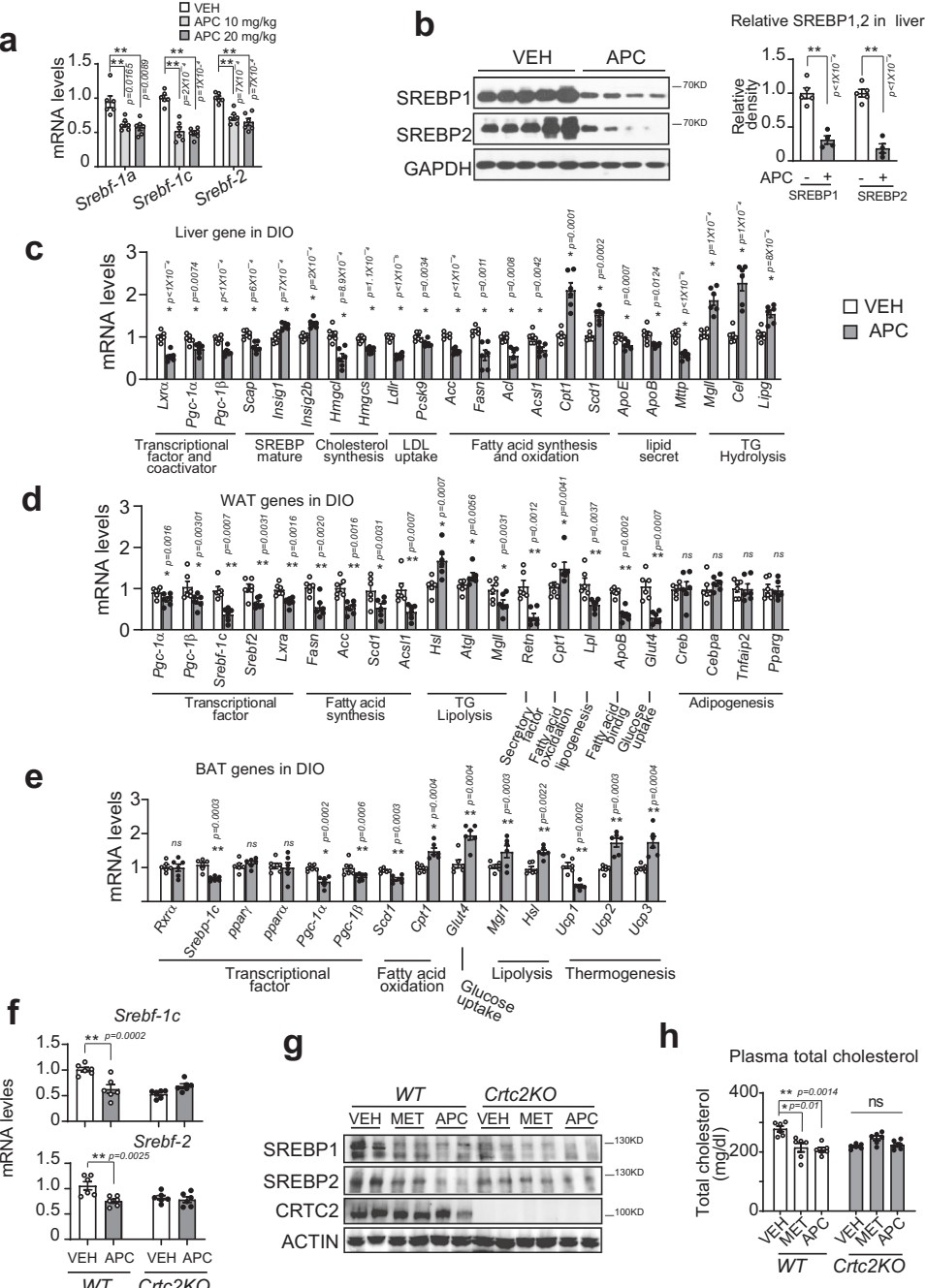

summary, these data suggested that APC ameliorates obesity through increasing insulin sensitivity.

**Developing novel CREB/CRTC2 inhibitors using APC as a lead compound.** As artepillin C (APC) displayed remarkable bioactivity to block CREB–CRTC2 protein interaction and enhance insulin sensitivity in vivo (Figs. 1–8 and Supplementary Figs. 1–8), we decided to design and discovery more potent inhibitor of CREB/CRTC2 (Fig. 9 and Supplementary Figs. 9 and 10 and Supplementary Table 4). We then examined the inhibitory activities of a series of in-house compounds with similar structures of APC by the CREB–CRTC2 two-hybrid system. Fortunately, compound **A32** was discovered with an IC$_{50}$ value of 9.95 μM, which displayed better inhibitory activity than APC. Then, based on the structure of APC and compound **A32**, a series

of novel compounds were designed and synthesized in order to increase the inhibitory activity of CREB/CRTC2 protein-protein interaction. (Fig. 9a, Supplementary Fig. 9, Supplementary Fig. 10a, and Supplementary Table 4). All the synthesized compounds were evaluated in vitro. Firstly, we reserved the 2-CF$_3$ group as R$^2$, R$^1$ substitution was explored. Amide compounds displayed better inhibitory activity than carboxylic acid and ester derivatives. Then, we reserved pharmacophore α, β-unsaturated amide, and assessed the effect of substitutions of the phenyl. Both the electron-donating and electron-withdrawing groups on the phenyl ring were tolerated for CREB/CRTC2 inhibition. The mono-substituted compounds demonstrated a regiochemical preference of ortho ≈ meta > para. When the R$^2$ group is 3-OMe, compound **A54** was afforded, the inhibitory activity of CREB/CRTC2 protein-protein interaction increased with an IC$_{50}$ value of 1.5 μM, which is 16-fold more potent than APC. Finally, when

**Fig. 6 SREBPs-mediated lipid metabolism is reduced by APC.** DIO mice were continuously i.p. injected with control vehicle (VEH) or artepillin C (APC) at indicated dose one time daily for 5 weeks ($n = 5–7$). **a** Quantitative PCR analysis of hepatic expressions of *Srebf-1a, -1c*, and *2* in these animals. Data are represented as mean ± SEM ($n = 6$ per group). *$P < 0.05$; **$P < 0.01$; $P$ values were determined by one-way ANOVA followed Dunnett's multiple comparisons test. **b** Immunoblotting analysis of hepatic SREBP-1c and 2 protein levels in these animals. One representative result from three independent experiments is shown (left), and relative SREBP1, and SREBP2 in the liver of these animals is normalized to GAPDH and is presented as a bar graph (right). Data are represented as mean ± SEM ($n = 6$ per group). *$P < 0.05$; **$P < 0.01$; $P$ values were determined by one-way ANOVA followed Dunnett's multiple comparisons test. **c** Quantitative PCR analysis of mRNA levels of genes related to lipid metabolism in the liver, (**d**) white adipose tissue (WTA), and (**e**) brown adipose tissue (BAT). Data are represented as mean ± SEM ($n = 6$). ns, $P > 0.05$; *$P < 0.05$; **$P < 0.01$; $P$ values were determined by unpaired two-tailed multiple $t$ test with two-stage linear step-up procedure, each gene was analyzed individually, without assuming a consistent SD. **f** Quantitative PCR analysis of mRNA levels of *Srebp-1c* and *Srebp-2* in the liver of *Crtc2*-KO mice and wild-type littermates ($n = 6$) fed with high-fat diet for 8 weeks. These animals were then orally administered with either vehicle control (VEH), or APC (20 mg/kg) one time daily for 3 weeks, followed by 8-h fasting before being sacrificed. One of two independent experiments is presented here. Data are represented as mean ± SEM. *$P < 0.05$; **$P < 0.01$; $P$ values were determined by the two-way ANOVA followed Bonferroni's multiple comparisons test. **g** SREBP1 and SREBP2 protein in the liver of *Crtc2*-KO mice ($n = 5$ mice per group, two technological repeats present a composite pool) as (**f**), and metformin (MET, 200 mg/kg) as a positive control. One of two independent experiments presented here, and relative SREBP1 and SREBP2 in the liver is shown as a bar graph (Supplementary Fig. 5a). **h** The plasma cholesterol of *Crtc2*-KO mice treated with MET and APC as (**g**). Data are represented as mean ± SEM ($n = 6$ per group). ns, $P > 0.05$; *$P < 0.05$; **$P < 0.01$; $P$ values were determined by two-way ANOVA followed Bonferroni's multiple comparisons test. Source data for this figure are provided as source data file.

$R^1$ group is replaced by (*S*)-4-phenyloxazolidin-2-one and $R^2$ group is 3-Ph, compound **A57** was obtained, which showed better capacity for inhibiting CREB/CRTC2 than APC in vitro (IC$_{50}$ 0.74 μM vs. 24.5 μM, respectively, Fig. 9b). Considering **A57** was a chiral compound, a racemic mixture **A1101** and another enantiomer **A58** were synthesized and their inhibitory activity in vitro was determined. Our results revealed that the inhibitory activity of **A57** (*S*-enantiomer) was higher than **A1101** (racemic) and **A58** (*R*-enantiomer) (Fig. 9c). These results suggest that the configuration of the phenyl group is related to the inhibitory activity. The configuration of compounds **A57** and **A58** is shown in Supplementary Fig. 9e. The different configuration of the phenyl group in these two compounds explains why compound **A57** has better inhibitory activity than **A58**. For the *S*-configured compound **A57**, the phenyl group in compound **A57** may occupy the active pocket of CREB protein, which increases the inhibitory activity. Consistent with the poorer inhibitory activity to CREB/CRTC2 interaction, compound **A58** exhibits a different configuration. It is possible that the steric bulk of the phenyl group in compound **A58** hinder interaction between compound **A58** and the protein CREB, which reduced the inhibitory activity. In addition, **A57** (cLog $P = 6.2$) was more potent than APC (cLog $P = 5.4$), due to the increased lipophilicity, and resulted in higher activity disrupting the CREB/CRTC2 interaction. Furthermore, treatment with 50 μM **A57** caused little damage to HEK293T cells (Supplementary Fig. 10b), excluding cellular toxicity from the inhibitory activity of **A57**. In common with APC, **A57** had little effect on CREB phosphorylation in primary hepatocytes (Supplementary Fig. 10c).

In vitro, **A57** showed a better capacity to reduce the mRNA levels of *G6pc* and *Pck1* than APC (10 μM) did in primary hepatocytes induced by glucagon (Fig. 9d). Correspondingly, oral administration of **A57** significantly decreased the fasting blood glucose levels in DIO mice, and *Crtc2*KO abolished these effects of **A57** and APC in vivo (Fig. 9e). It was suggested that liver-specific CRTC2 knockout improves whole-body energy metabolism by increasing FGF21 expression through CRE-miR34a-FGF21 axis[24]. Consistently, APC and **A57** increased plasma and liver FGF21 in a wild-type control group but did not further increase them in *Crtc2*KO mice, in which the basal levels of plasma and liver FGF21 have been dramatically raised by *Crtc2*KO (Supplementary Fig. 10f). Next, we tested the pharmacological effects of **A57** in *db/db* mice. Expectedly, one oral delivery of **A57** (5 mg/kg) significantly reduced fasting blood glucose levels in these animals, whereas APC exhibited little effect with the same dosage (Fig. 9f). Correspondingly, **A57** showed

stronger bioactivity to improve oral glucose tolerance (OGTT) than APC in *db/db* mice with daily oral administration for 3 weeks (Fig. 9g).

We further investigated the bioavailability of **A57** and APC administrated orally (p.o.) or intravenously (i.v.) to ICR/JCL mice. The results (Table 1) showed that compound **A57** displayed a good AUC (648 ng·h/mL) and better oral bioavailability (25.3% vs. 14.4%) as compared to APC. These data indicated that more effective bioactivity of **A57** was tightly associated with increased bioavailability. Collectively, the above data suggested that this series of compounds could serve as possible lead compounds for the development of CREB/CRTC2 inhibitors.

## Discussion

The glucagon-cAMP-CREB/CRTC2 signaling axis is essential for the control of fasting blood glucose levels[18]. In this study, a cell-based assay detecting CREB–CRTC2 interaction was developed to screen individual compounds from Brazil green propolis, which identified APC acts as a small-molecule inhibitor of the CREB/CRTC2 transcriptional complex, leading to improved insulin sensitivity in obsess mice induced by diet or genetic mutation (DIO or *db/db*, respectively). Surprisingly, APC has also been found to reduce the transcription of SREBPs and their target genes, and to increase plasma FGF21 levels, which leads to the improvement of hyperlipidemia in these animals. These metabolic benefits of APC contribute to increases in the insulin sensitivity of diabetic mice and prompted us to create more effective inhibitors than APC. Optimizing a compound based on APC, we designed and generated a series of small molecules. Among them, we identified a novel α,β-unsaturated amide compound (**A57**) that showed higher inhibitory activity and better oral bioavailability to improve insulin sensitivity in *db/db* mice than APC. In summary, the natural propolis-derived compound APC, an inhibitor of CREB/CRTC2, leads us to create a new compound (**A57**) for developing novel anti-diabetic drugs.

The cAMP-CREB signaling pathway has long been known to play a central role in adipocyte differentiation[50,51]. Moreover, collective analysis of CREB-CRTC2 function in peripheral tissues suggests that chronic activation of this pathway contributes to pathological changes associated with insulin resistance, including hyperglycemia, hyperinsulinemia, and adipose tissue inflammation[52–55]. Thus, small molecules that effectively reduce CREB/CRTC2 activity, particularly in the liver and adipose tissues, may provide therapeutic benefits to the affected individuals, which are validated by the findings with APC and **A57** in this

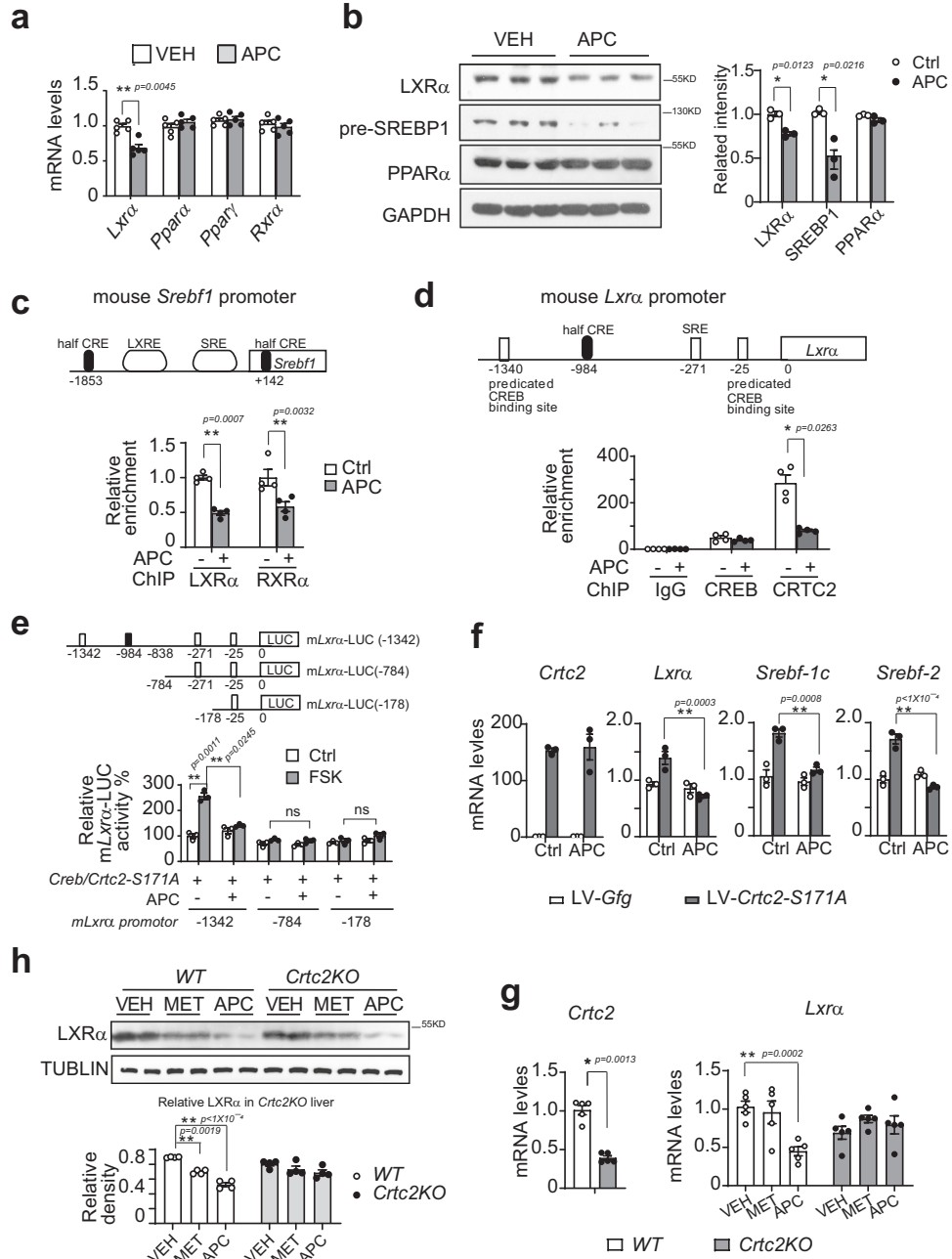

study. By binding with CREB, inhibitors disrupt CREB/CRTC2 complex formation, resulting in the reduction of hepatic gluconeogenic gene expression (*Pgc-1α*, *Pck1*, *G6pc*, etc. Fig. 6) and fasting blood glucose levels (Fig. 8). The subsequent decreases in the coactivators PGC-1α/β and the nuclear receptor LXRα protein levels further reduce the transcription of the master lipid regulators, SREBP1 and SREBP2, as well as their target genes (*Fasn*, *Acc*, and *Ldlr*, etc.), leading to the attenuation of circulating TG and cholesterol levels by inhibiting lipid biosynthesis in the liver (Figs. 5–7). At the molecular level associated with APC-mediated reduction of SREBP1 protein, several half CRE sites were also found in the *Srebf1* promoter (Fig. 7c, up). Moreover, these half CRE sites directly regulate the transcription of SREBP2[25], which was confirmed by this work. On the one hand, the overexpression of CRTC2-S171A, a constantly activated CRTC2, significantly induced the mRNA levels of *Srebf-1c* and *Srebf2* (Fig. 7f). In addition, *Crtc2*KO attenuated APC further reducing the protein

levels of SREBP1 and 2 in the liver (Fig. 6g). Therefore, we demonstrated that CRE played a critical role in controlling the transcript of both *Srebf-1a, -1c* and *-2*, which was the critical molecular mechanism of an inhibitor of CREB/CRTC2 interaction. Moreover, we would like to suggest that APC and **A57** were special in reducing hepatic CREB/CRTC2 function, bringing benefits not only for glucose but also lipid metabolism.

CREB has been shown to interact with its coactivators via different domains, for example, the KID domain with CBP/P300[56,57] and the ZIP domain with CRTCs[58]. The results of both ITC and SPR assays indicated that APC directly bound with CREB protein (Fig. 3a and Supplementary Fig. 2d), by which APC disrupted CREB interaction with CRTC2. Our data further suggested that the CREB-ZIP domain might be involved in APC binding with CREB (Fig. 3c). Although APC had little effect on the interaction between CREB and CBP/P300 (Fig. 2c and Supplementary Fig. 2a), it significantly reduced CBP-mediated

**Fig. 7 APC inhibits SREBP1 expression via the CRE-LXRα-LXRE axis. a** Quantitative PCR analysis of hepatic expressions of *Lxra, Ppra, Pprg, Rxra* in DIO mice, which were continuously i.p. injected with control vehicle (VEH) or APC (20 mg/kg) with indicated doses one time daily for 5 weeks ($n = 6$ per group). One of two independent experiments is shown here. Data are represented as mean ± SEM. *$P < 0.05$; **$P < 0.01$; $P$ values were determined by two-way ANOVA followed Bonferroni's multiple comparisons test. **b** Immunoblotting analysis of hepatic LXRα and premature SREBP1 protein levels in the livers of DIO mice injected APC for 5 weeks as (**a**). One of three independent experiments is shown here (left). The relative protein of LXRα, SREBP1, and PPARα normalized by GAPDH is shown as a bar graph (right) ($n = 3$). Data are represented as mean ± SEM. *$P < 0.05$; **$P < 0.01$; $P$ values were determined by two-way ANOVA followed Bonferroni's multiple comparisons test. **c** CHIP analysis *LXRa* and *RXRa* recruited to the promoter of the mouse *Srebf1* (top) gene in primary hepatocytes from wild C57 mice and pretreated APC (10 μM) for 1-h (bottom) ($n = 4$ per treatment). One of three independent experiments presented here. Data are represented as mean ± SEM. *$P < 0.05$; **$P < 0.01$; $P$ values were determined by two-way ANOVA followed Bonferroni multiple comparisons test. **d** CHIP analysis of CREB and CRTC2 occupancy over the half CRE sites in the *Lxra* promoter (Top) in primary mouse hepatocytes overexpressing CRTC2, pretreated with APC (10 μM) for 1-h prior to 1-h glucagon stimulation (Gluc, 100 nM) (bottom) ($n = 4$ per treatment). One of three independent experiments presented here. Data are represented as mean ± SEM. *$P < 0.05$; **$P < 0.01$; $P$ values were determined by two-way ANOVA followed Bonferroni's multiple comparisons test. **e** Reporter activity of truncated *Lxra-LUC* reporter. The luciferase reporter was driven by mouse *Lxra* promoter (−1342) or truncations (−784 and −178) co-transfected with *MYC-Creb* and *FLAG-Crtc2-S171A* in HEK293T cells, and pretreated with APC prior to stimulation with FSK (10 nM) for 4-h ($n = 3$ per treatment). One of three independent experiments presented here. Data are represented as mean ± SEM. *$P < 0.05$; **$P < 0.01$; $P$ values were determined by two-way ANOVA followed Bonferroni's multiple comparisons test. **f** Quantitative PCR analysis of the expression of *CRTC2, Lxra, Srebf-1c,* and *Srebf-2,* in primary hepatocytes, which were infected by LV-*Crtc2-S171A* or control and incubated with APC (10 μM) for 8-h ($n = 3$ per treatment). One of three independent experiments presented here. Data are represented as mean ± SEM. *$P < 0.05$; **$P < 0.01$; $P$ values were determined by two-way ANOVA followed Bonferroni's multiple comparisons test. **g** Quantitative PCR analysis of *Lxra* gene expression in the liver of CRTC2-KO mice. Be induced by high-fat diet for 8 weeks, with *Crtc2*-KO mice and wild-type littermates administered VEH (vehicle control), MET (metformin, 200 mg/kg), or APC (20 mg/kg) for 3 weeks, followed by fasting 8-h before anesthesia ($n = 5$ per group). Data are represented as mean ± SEM. *$P < 0.05$; **$P < 0.01$; $P$ values of *Crtc2* gene were determined by unpaired two-tailed $t$ test with Welch's correction, and $P$ values of *Lxra* gene were determined by two-way ANOVA followed Bonferroni's multiple comparisons test. **h** Immunoblotting analysis of LXRα protein in the liver of these mice as in (**g**). One of two independent experiments is presented here (top), and the relative concentration of LXRα normalized to ACTIN is shown as a bar graph (bottom). ($n = 5$ mice per group, two technological repeats represent a composite pool). Data are represented as mean ± SEM. *$P < 0.05$; **$P < 0.01$; $P$ values were determined by two-way ANOVA followed Bonferroni's multiple comparison test. Source data for this figure are provided as source data file.

histone acetylation (Fig. 3d). These results demonstrated that the formation of the CREB/CRTC2 complex was crucial for the recruited CBP/P300 activation, whereas, whether other coactivators were involved in the inhibition by APC warrants further exploration in the future.

As an important transcriptional coactivator, PGC-1α increases insulin resistance through strong activation of PPARα dependent transcription in the liver[59] and increasing PPARγ-mediated lipid syntheses in adipose tissues[60]. Therefore, reducing CREB-mediated PGC-1α expression may benefit the treatment of metabolic disorders, which was further verified in this study. Meanwhile, the inhibitory effect of APC on SREBP1 expression was further amplified by the reduction of LXRα transcription (Fig. 7), as the LXRα/PPARα complex positively activates *Srebp-1c* transcription by binding with LXRE sites[41,61]. In contrast to statins that lower LDL cholesterol and triglycerides but activate SREBP1 expression[62,63], APC reduces serum cholesterol levels, as well as the sizes of lipid droplets in the liver (Fig. 5), which was tightly associated with strongly decreasing SREBP expression. Therefore, APC was capable of improving lipid homeostasis by inhibiting SREBP function, which avoids the side effects of the fatty liver associated with other treatments. Although a previous study reported that APC enhances 3T3-L1 pre-adipocyte differentiation as a weak PPARγ ligand[64], the evidence that APC reduced the fat to lean ratio as shown in this study (Fig. 5b and Supplementary Fig. 4c) suggested that PPARγ may not be the primary target of APC in vivo, since thiazolidinediones, the classic PPARγ ligands, increase white adipose tissue sizes in animals[65,66]. APC decrease of body weight and plasma leptin levels in *db/db* mice were associated with it increasing insulin sensitivity and RER (Fig.8), which brings benefits for improving whole-body metabolism.

Oral Brazilian green propolis significantly reduced a *G6P*-luc reporter in the liver of DIO mice (Fig. 1b) and remarkably decreased fasting blood glucose in *db/db* mice (Supplementary Fig. 1a), which demonstrated that Brazilian green propolis improves insulin sensitivity in diabetic mice at least partially through inhibiting CREB-mediated hepatic gluconeogenesis. Targeting gluconeogenic enzymes or their regulators in the liver is a strategy for type 2 diabetes treatment, because it is generally accepted that enhanced gluconeogenesis is the major contributor to the increased blood glucose levels observed in diabetic patients[67–69]. The propolis-mediated lowering of blood glucose indicated that some natural inhibitor of CREB-mediated gluconeogenesis was pooled in propolis. Besides activating AMPK kinase, metformin has also been reported to inhibit the activation of PKA and decrease cAMP levels[70]. Unlike metformin, unexpectedly, Brazilian green propolis and the propolis-enriched compound APC had little effect on the phosphorylation of CREB and cAMP accumulation induced by glucagon (Figs. 1d and 4d and Supplementary Fig 10c). Moreover, no impacts of propolis on CRTC2 nuclear-shifting was detected (Supplementary Fig. 3d), which suggested that propolis and APC target the events after the occurrence of CREB phosphorylation in the glucagon-CREB/CRTC2 signaling pathway, and demonstrated that the effects of propolis and APC were limited to the end of the PKA-CREB/CRTC2 pathway.

Due to multiple factors including honeybee species, botanical source, season, and extraction method[71–73], there is a huge variation in the chemical composition of propolis, which greatly influences its pharmacological efficacy on patients with type 2 diabetes. Thus, further research is required to identify some novel anti-diabetic components from propolis to guide new drug discovery. Although numerous efforts have been made in this regard, it remains controversial which compounds contribute to such beneficial property of propolis, mainly as a result of the lack of knowledge of the molecular targets of propolis. In this study, APC was found to play a major role in Brazilian green propolis-mediated amelioration of hyperglycemia and hyperlipidemia in diabetic mice, which depended on its ability to disrupt CREB–CRTC2 interactions.

The functional diversity of CRTC2 is critical for maintaining whole-body metabolism balance. As a coactivator, CRTC2

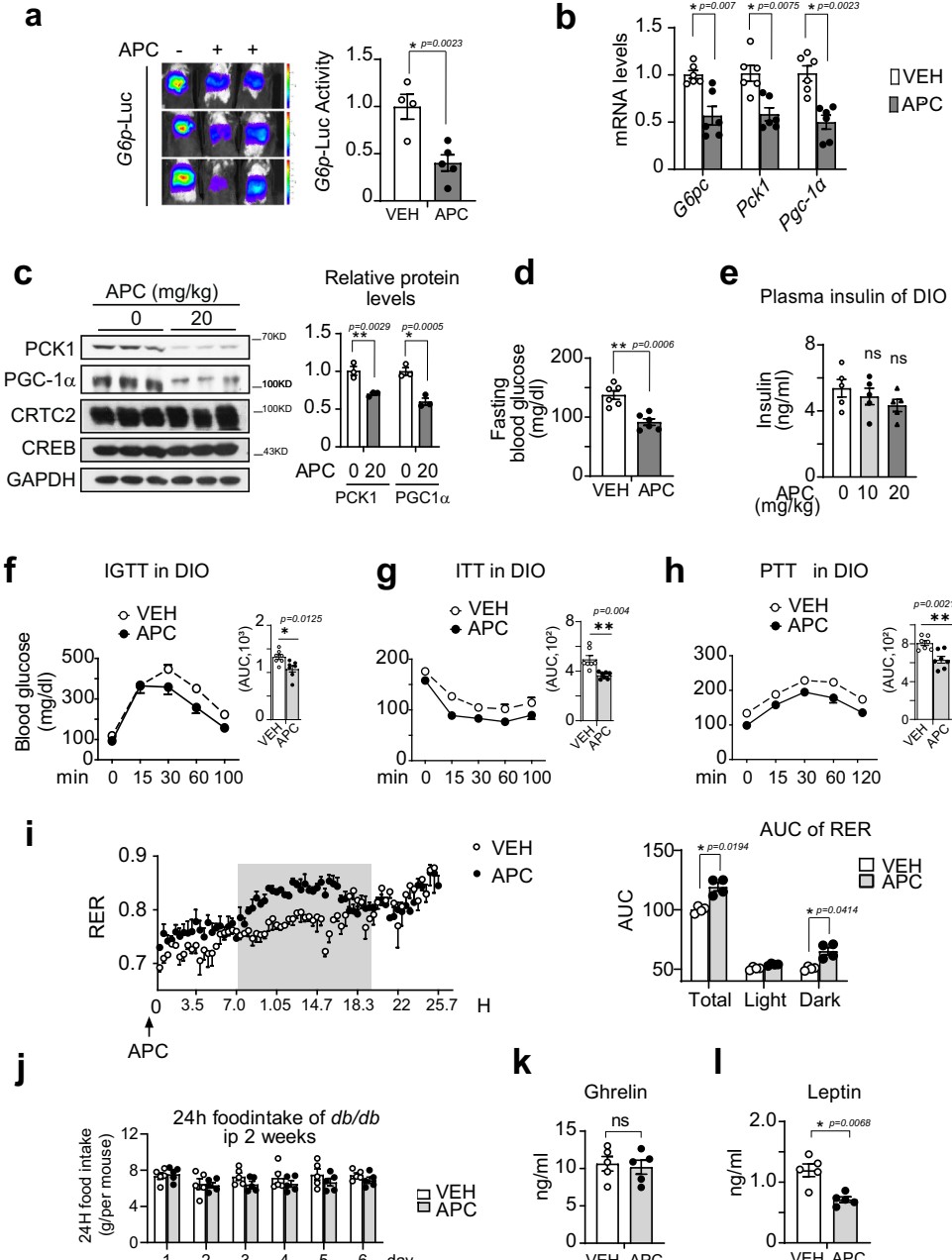

effectively enhances PKA-CREB-CRE-regulated gluconeogenesis in the liver. Several works have revealed that prolonged activation of CRTC2 under insulin resistance might play a critical role in hyperglycemia[19,20,74]. In contrast, mTORC1 suppresses COX-2 expression by phosphorylating CRTC2 at Ser136, leading to enhancement of white adipocyte browning[75]. Moreover, it has been reported that the transcriptional function of CRTC2 is necessary for the secretion of insulin and GLP-1 in pancreatic β cells[30] and intestinal L cells[29], respectively. Therefore, tissue-specific inhibition of CRTC2 transcription is more pharmacological in practice. In this work, the liver-specific enrichment of APC (Fig. 5k, l) greatly aided in maintaining a stable and relative higher concentration in the liver, where APC reduced hepatic glucose and lipid production specifically.

In addition, bio-distribution of APC to the liver also reduced the side effects in other tissues, for example, pancreatic β cells and intestinal L cells. Recently, hepatic CRTC2 has been shown to control whole-body energy metabolism via the CRE-miR-34a-

FGF21 axis[24]. Our work confirmed this pathway and demonstrated that compounds APC and **A57** could mimic liver CRTC2 depletion to increase liver and plasma FGF21 levels, which further improved whole-body energy metabolism. Therefore, our work suggested that APC and **A57** could improve energy metabolism through multiple CREB/CRTC2-mediated pathways in the liver.

mTORC1 has been reported to enhance SREBP1c-mediated lipogenesis mainly by increasing the maturation of SREBP1c proteins[26]. Our previous study has demonstrated an inhibitory function of CREB/CRTC2 in the reduction of hepatic mTORC1 activity in fasted mice via increasing the expression of PER2[76]. These works could lead to a scenario in which APC would decrease CREB/CRTC2-mediated expression of PER2 to cause the enhancement of mTORC1 activity and subsequent increase of SREBP1c activation that is inconsistent with our data in this study. We performed preliminary research and found that APC reduced CREB/CRTC2-mediated transcription of PER2, increased mTORC1 activity, and decreased the protein amounts

**Fig. 8 APC increases insulin sensitivity in diabetic mice. a** Live imaging assay of *G6P-Luc* in DIO mice i.p. administered APC (20 mg/kg) or vehicle (VEH) one time daily for 3 days and then fasted for 16-h before imaging. Hepatic Ad-*G6P-Luc* activity is normalized to *β-Gal* activity ($n = 5$ per group, three mice shown here). One of two independent experiments is shown here. Data are represented as mean ± SEM. *$P < 0.05$; **$P < 0.01$; $P$ values were determined by unpaired two-tailed $t$ test with Welch's correction. DIO mice were continuously i.p. injected with a control vehicle (VEH) or APC (20 mg/kg) one time daily for 5 weeks ($n = 5–7$). **b** Quantitative PCR analysis of *G6pc*, *Pck1*, and *Pgc-1α* gene expression in the liver of these DIO mice. One of two independent experiments is shown here ($n = 6$ per group). Data are represented as mean ± SEM. *$P < 0.05$; **$P < 0.01$; $P$ values were determined by unpaired two-tailed multiple $t$ test with the two-stage linear step-up procedure, each gene was analyzed individually, without assuming a consistent SD. **c** Immunoblotting analysis of endogenous protein levels of PCK1, PGC-1α, and CRTC2 in the livers of 16-h fasted DIO mice i.p. treated with APC (20 mg/kg) or vehicle (VEH) one time daily for 5 weeks ($n = 6$ per group, three repeats represented pooling sample). One of two independent experiments is shown here. Data are represented as mean ± SEM. *$P < 0.05$; **$P < 0.01$; $P$ values were determined by two-way ANOVA followed Bonferroni's multiple comparisons test. **d** Measurement of blood glucose levels in 16-h fasted DIO mice i.p. treated with APC (20 mg/kg) or vehicle (VEH) one time daily for 2 weeks ($n = 6$ per group). One of three independent experiments is shown here. Data are represented as mean ± SEM. *$P < 0.05$; **$P < 0.01$; $P$ values were determined by unpaired two-tailed $t$ test with Welch's correction. **e** The plasma insulin levels in DIO mice i.p. administered APC at indicated concentrations or vehicle (VEH) one time daily for 5 weeks ($n = 5$ per group). One of three independent experiments is shown here. Data are represented as mean ± SEM. ns, $P > 0.05$; *$P < 0.05$; **$P < 0.01$; $P$ values were determined by one-way ANOVA followed Dunnett's multiple comparisons test. **f** Injected glucose-tolerance test (IGTT), **g** insulin-tolerance test (ITT), and **h** pyruvate tolerance test (PTT) of DIO mice i.p. administered APC (20 mg/kg) or vehicle (VEH) one time daily for 5 weeks ($n = 7$ per group). Area under curve (AUC) analysis of each test is shown as a bar graph in the upper-right corner of each curve. One of two independent experiments is shown here. Data are represented as mean ± SEM. *$P < 0.05$; **$P < 0.01$; $P$ values of the curve were determined by two-way ANOVA followed Bonferroni's multiple comparisons test, and $P$ values of AUC were determined by unpaired two-tailed $t$ test with Welch's correction. **i** Measurement of respiratory exchange ratio (RER) in these mice ($n = 4$ per group) after 2 weeks of APC injection. The time courses of RER curves (left) were analyzed by AUC measurement (right). The shadow areas indicate dark times. One of two independent experiments is shown here. Data are represented as mean ± SEM. *$P < 0.05$; **$P < 0.01$; $P$ values of curve and RER AUC were determined by two-way ANOVA followed Bonferroni's multiple comparisons test. **j** The daily food intake of *db/db* mice after 2 weeks of APC injection ($n = 5$ per group). One of two independent experiments is shown here. Data are represented as mean ± SEM. ns, $P > 0.05$; *$P < 0.05$; **$P < 0.01$; $P$ values were determined by two-way ANOVA followed Bonferroni's multiple comparisons test. **k** Measure of plasma ghrelin and leptin (**l**) levels of *db/db* mice after 7 weeks of APC injection. One of two independent experiments is shown here. Data are represented as mean ± SEM ($n = 5$ per group). ns, $P > 0.05$; *$P < 0.05$; **$P < 0.01$; $P$ values were determined by unpaired two-tailed $t$ test with Welch's correction. Source data for this figure are provided as source data file.

of mature SREBP1 and fat acid synthase in glucagon-insulin-treated primary hepatocytes, which suggests that APC is capable of reducing lipogenic gene expression when it increases mTORC1 activity at the same time. Interestingly, a recent paper has demonstrated that mTORC1 hyperactivation caused by liver-specific TSC1 deletion is insufficient to induce SREBP1c activation and de novo lipogenesis in the liver of high-fat diet-fed mice[77]. Considering the requirement of TSC1 for PER2 to inhibit mTORC1 activity[76], the reduction of PER2 by APC may not be sufficient to induce SREBP1c activation. Moreover, our data suggest that APC reduces the transcription of SREBP1c in vivo (Fig. 6b, g), which would result in the decrease of whole SREBP1c protein levels and overcome the effect on lipogenesis caused by APC activation of mTORC1.

Although APC remarkably increased insulin sensitivity in diabetic mice, it displayed the low inhibitory activity of CREB/CRTC2 interaction ($IC_{50} = 24.5$ μM). Aiming to find a more potent and druggable inhibitor, a systematic structural modification of APC was conducted. All the synthesized compounds were evaluated in vitro. Mono-substituted compounds demonstrated the regiochemical preference ortho ≈ meta > para. Both electron-donating and electron-withdrawing groups on the phenyl ring were tolerated for CREB/CRTC2 inhibition. Among them, compounds with 3-phenyl substitution exhibited good inhibitory activity against CREB/CRTC2 interaction. Specially, the $R^1$ substitution was explored herein. Amide compounds displayed better inhibitory activity than carboxylic acid derivatives. A novel α,β-unsaturated amide (**A57**) was created, which displayed 30 times higher inhibitory activity against CREB/CRTC2 protein interaction ($IC_{50} = 0.74$ μM) than the natural product APC. Associated with more potent lipophilicity and bioavailability than APC, compound **A57** also displayed a significant reduction of both fasting blood glucose and postprandial glucose in *db/db* mice. Moreover, an α, β-unsaturated amide was revealed as the pharmacophore of the CREB/CRTC2 protein-interaction inhibitor. This novel scaffold of CREB/CRTC2

protein-interaction inhibitor demonstrated a promising avenue for the discovery and development of new anti-diabetic drug candidates.

In summary, we identified APC as a specific small-molecule inhibitor of the interaction between CREB and CRTC2 from the traditional medicine propolis. Based on inhibiting CRE transcription, APC reduced the levels of blood glucose and lipids by inhibiting gluconeogenesis and lipid synthesis, respectively. APC's ability to increase insulin sensitivity prompted us to design a stronger inhibitor (**A57**). This work revealed a mechanism of CREB/CRTC2 that translated into the creation of the novel compound **A57**, a promising candidate for a new drug to combat type 2 diabetes.

## Methods

**Plasmids and adenoviruses.** Complete DNA sequences encoding human *Creb* and *Crtc2* were cloned in the plasmids psc811 and psc804 to express BD-CREB and VP16AD-CRTC2 recombind proteins, respectively. *GAL4-luciferase* and RSV (Rous Sarcoma Virus)-*Renilla* were constructed using the pGL3 plasmid. pGEX5X-1 and pET28C were used to construct plasmids for GST- and HIS-tagged proteins expression, respectively. Adenoviruses encoding Ad-HA-*Crtc2*, Ad-*GFP*, and Ad-RSV-*β-Gal* (galactosidase) have been described previously[78]. For infecting primary hepatocytes, 0.5–1 MOI adenovirus was employed. For live imaging experiments, $10^8$ plaques forming units (*pfu*) Ad-*G6P-Luc* and $5 \times 10^7$ *pfu* Ad-*RSV-β-gal* adenovirus were delivered to mice by tail vein injection. Mice were then imaged on day 3–5 after adenovirus delivery.

**Cell culture and luciferase reporter assays.** HEK293T cells were cultured in DMEM containing 10% FBS as well as 1% penicillin and streptomycin (Sigma) and transfected using PEI (Sigma) for transient expression or reporter assays. Mouse primary hepatocytes were isolated, cultured, and infected with adenoviruses as described previously[19]. Briefly, the mouse liver was perfused with washing buffer and followed by digestion buffer supplemented with collagenase II (Sigma). The digested liver tissue was excised, dispersed, and scraped in washing buffer, and hepatic parenchymal cells were filtered through a 100 μM mesh and pelleted in M199 media. These were washed two times, then the isolated primary hepatocytes were seeded to collagen-coated plates or dishes. After incubation with M199 supplied FBS for 4–6 h, hepatocytes were infected with adenoviruses (0.5–1 MOI) for 12-h before experiments.

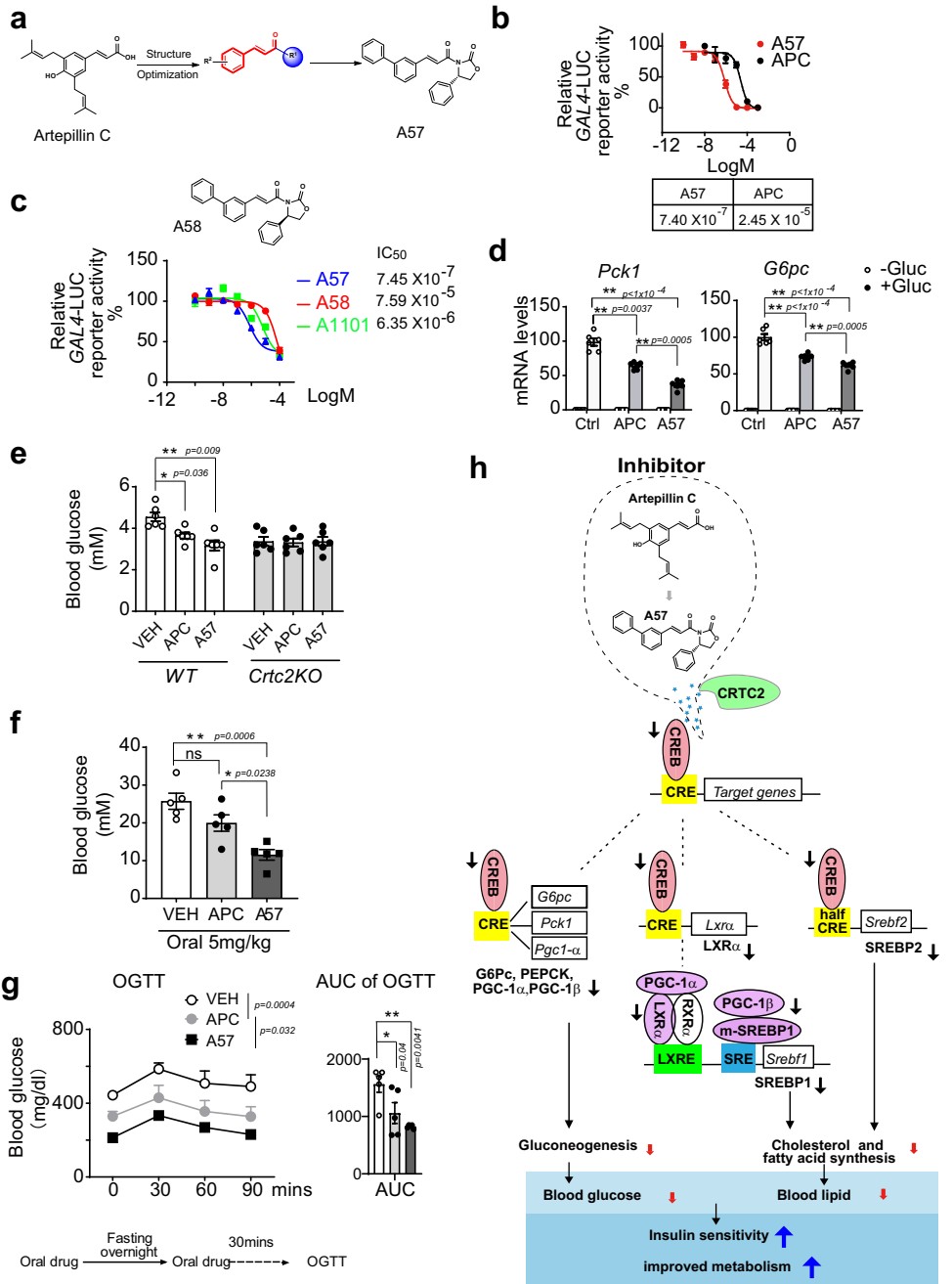

**Luciferase enzymatic assays**. HEK293T cells ($4 \times 10^5$ per well of 24-well-plate) were transfected with plasmids expressing *CRE-luciferase* reporter (50 ng) together with *RSV-β-gal* (20 ng) by PEI-media overnight and induced by FSK (forskolin 10 nM) for 6-h. The cells were then collected in the lysis buffer (25 mM Gly-gly, 15 mM MgSO$_4$, 4 mM EGTA, 1% V/V Triton X-100, 1 mM DTT, pH 7.8), mixed with an equal volume of luciferase substrate buffer (25 mM Gly-gly, 15 mM MgSO$_4$, 4 mM EGTA, 250 mM potassium phosphate, 2.5 mM ATP, 2 mM DTT, 5 mg/mL luciferin, pH 7.8), and subjected to EnVision Multimode Plate Reader (Perkin/Elmer) to determine generated fluorescent signals.

**Bioassay-guided Isolation**. First, 500 mL of Brazilian green propolis was concentrated in a vacuum to give a dark residue (128.4 g), which was resuspended in H$_2$O (3 L) and successively partitioned into petroleum ether (10.9 g), EtOAc (90.8 g), *n*-BuOH (12.7 g), and water-soluble fractions (13.7 g, Supplementary Fig. 1f, g). Using a *G6P-Luc* reporter assay in primary hepatocytes, the EtOAc soluble fraction showed the most significant inhibition (Supplementary Fig. 1f). Then, this fraction was subjected to MCI Gel CHP20P column chromatography (MeOH/H$_2$O, 40–100%) to afford 40% (3.2 g), 60% (21.8 g), 80% (37.2 g), and 100% (29.6 g) MeOH-eluted subfractions that were further purified by silica gel, Sephadex LH-20 (MeOH), and preparative HPLC column chromatography. 40%

MeOH-elution yielded subfraction P1 (162 mg) and P9 (16 mg). 60% MeOH-elution gave subfractions P2 (471 mg), P4 (156 mg), P7 (498 mg), P10 (56 mg), P12 (125 mg), P14 (25 mg), P15 (87 mg), P17 (29 mg), P20 (29 mg), and P21 (66 mg). 80% MeOH-elution yielded subfraction P3 (391 mg), P5 (10 mg), P6 (20 mg), P11 (15 mg), P13 (1005 mg), P16 (867 mg), P18 (62 mg), and P19 (134 mg). 100% MeOH-elution led to subfraction P8 (274 mg). P1–P6 were identified as (*E*)-*p*-coumaric acid (P1), drupanin (P2), artepillin C (P3), (*E*)-*p*-hydroxycinnamic acid ethyl ester (P4), (*E*)-*p*-coumaric acid benzyl ester (P5), and (*E*)-2,3-dihy-droconiferyl-*p*-coumarate (P6). P7-P21 were capillartemisin A (P7), baccharin (P8), (*E*)-caffeic acid (P9), (*E*)-caffeic acid ethyl ester (P10), 2,2-dimethylchromene-6-propenoic acid (P11), kaempferol (P12), kaempferide (P13), quercetin (P14), 6-methoxykaempferol (P15), betuletol (P16), 5,6,7,-trihydroxy-3,4′-dime-thoxyflavone (P17), pinocembrin (P18), isosakinanetine (P19), aromadendrin (P20), and dihydrokaempferide (P21), and all fractions were subjected to detailed NMR and MS analysis.

**Synthesis of artepillin C**. For in vivo experiments, 5.0 g of APC was synthesized using a method described previously[79]. The obtained APC was >98% pure, and confirmed by HPLC and NMR. The standard for APC was purchased from Wako Pure Chemical Industries Limited (Japan).

**Fig. 9 A57, a novel inhibitor targets CREB/CRTC2 interaction. a** Structure of APC (left), the reserved α,β-unsaturated ketone group (middle), and compound **A57** (right). **b** $IC_{50}$ of **A57** and APC determined using a CREB/CRTC2 two-hybrid reporter system. One of three independent experiments presented here. Data are represented as mean ± SEM ($n = 3$ per treatment). $IC_{50}$ determined by nonlinear regression curve fit in Graphpad 8.0. **c** $IC_{50}$ of compound **A57**, **A58** (top) and racemic **A1011** determined by CREB/CRTC2 two-hybrid system (bottom). The compounds **A57**, **A58** and **A1011** were incubated with cells for 12-h. One of three independent experiments presented here. Data are represented as mean ± SEM ($n = 3$ per treatment). $IC_{50}$ determined by nonlinear regression curve fit in Graphpad 8.0. **d** mRNA level of gluconeogenesis marker, *Pck1* and *G6pc* in primary hepatocytes incubated with APC or **A57** (10 mM) 1-h prior to glucagon (100 nM) stimulation for 4-h ($n = 6$ per treatment). One of three independent experiments presented here. Data are represented as mean ± SEM. *$P < 0.05$; **$P < 0.01$; $P$ values were determined by two-way ANOVA followed Bonferroni's multiple comparisons test. **e** Measure of 16-h fasting blood glucose in *Crtc2*KO mice and wild-type littermates induced by high-fat diet (DIO) for 8 weeks and oral APC, **A57** (20 mg/kg) or vehicle (VEH) daily for 3 days ($n = 6$ per group). One of three independent experiments presented here. Data are represented as mean ± SEM. *$P < 0.05$; **$P < 0.01$; $P$ values were determined by two-way ANOVA followed Bonferroni's multiple comparisons test. The *db/db* mice were orally treated with **A57** (5 mg/kg), APC (5 mg/kg), or vehicle (VEH), one dose per day for 3 weeks. **f** Measure of 16-h fasting blood glucose levels in these *db/db* mice after one oral administration of compounds ($n = 5$ per group). One of three independent experiments presented here. Data are represented as mean ± SEM. ns, $P > 0.05$; *$P < 0.05$; **$P < 0.01$; $P$ values were determined by one-way ANOVA followed Turkey's multiple comparisons test. **g** Oral glucose-tolerance test (OGTT) in these *db/db* mice administrated oral drugs for 3 weeks (left) ($n = 5$ per group). Area under curve (AUC) analysis is shown as a bar graph (right). One of three independent experiments presented here. Data are represented as mean ± SEM. *$P < 0.05$; **$P < 0.01$; $P$ values were determined by two-way ANOVA followed Bonferroni's multiple comparisons test. **h** The molecular mechanism of how these small molecules, APC and **A57**, improve metabolic syndrome by disrupting the interaction between CREB and CRTC2. Source data for this figure are provided as a Source data file.

**Table 1 Pharmacokinetic properties of compounds A57 and APC in ICR mice.**

| Compd | Admin. | Dose (mg/kg) | $C_{max}$ (ng/mL) | $T_{max}$ (h) | $T_{1/2}$ (h) | $AUC_{0-t}$ (ng·h/mL) | MRT (h) | Cl (L/h/kg) | $F_{po}$ (%) |
|-------|--------|--------------|-------------------|---------------|---------------|------------------------|---------|-------------|--------------|
| A57 | p.o. | 20 | 165 | 0.67 | 2.11 | 648 | 3.43 | – | 25.3 |
| | i.v. | 2 | – | – | 1.21 | 256 | 1.40 | 131 | – |
| APC | p.o. | 20 | 510 | 0.25 | 3.29 | 616 | 4.33 | – | 14.4 |
| | i.v. | 2 | – | – | 5.89 | 428 | 2.46 | 78 | – |

$C_{max}$ peak plasma concentration, $T_{max}$ time to peak plasma concentration, $T_{1/2}$ elimination half-life, $AUC_{0-t}$ area under the plasma concentration-time curve, MRT mean residence time, Cl clearance, $F_{po}$ the relative bioavailability after oral administration.

**Mammalian two-hybrid assay.** HEK293T cells were seeded in 24-well plates at a density of $5 \times 10^4$ cells/well for 12-h, followed by PEI-mediated co-transfection with reporter plasmids (50 ng *Gal4-Luc* and 25 ng *RSV-Renilla*), plasmids of VP16AD-*Crtc2* wild type or S171A (100 ng) and *BD-Creb* (100 ng) per well. After 5 h of transfection, cells were exposed to compounds or vehicle (DMSO) for 16 h. Cell extracts were prepared and subjected to luciferase assays as described previously[17]. The luciferase activity was normalized to that of Renilla.

**Animals and experimental design.** All animal experiments were performed according to procedures approved by the Shanghai Institutes for Biological Sciences (SIBS) Animal Care and Use Committee. The *db/db* mice (stock NO. 000642 C57BL/6J, JACKSON) were purchased from the Jackson laboratory. The 8–10-week-old wild-type C57BL/6 and ICR/JCL mice were purchased from the Shanghai Laboratory Animal Center. Mice were housed in the animal facility at SIBS, in which mice were maintained on a 12-h light/12-h dark cycle for at least 2 weeks before the study and had free access to water and a regular diet (12% fat). For high-fat-induced obesity mice (DIO), high-fat diet (60% fat, Research Diets) was fed to wild-type mice for at least 13–15 weeks and glucose-tolerance tests were performed to confirm the development of diabetes in these mice. After being euthanized, samples were collected. Unless specified, male mice were used for these experiments.

**Living NMR assay.** Under the condition of non-anesthesia and non-invasion, the composition of fat and lean in live mice was analyzed by nuclear magnetic resonance (NMR, The Minispec, Bruker) according to the operational manual.

**Live imaging analysis.** After two days of tail intravenously delivery of Ad-*G6p-Luc* ($10^8$ *pfu*) and Ad-*RSV-β-Gal* ($5 \times 10^7$ *pfu*) adenoviruses, mice were orally administered propolis or i.p. injected with APC one dose per day for another 3 days. After 16 h fasting, mice were anesthetized using gaseous isoflurane (Abbott), were i.p. injected with 100 mg/kg sterile firefly D-luciferin (Biosynth AG), and 15 min later, imaged by IVIS-100 Imaging System and analyzed with Living Image Software (Xenogen). Liver lysates were prepared to determine β-Gal activity. Luciferase activity detected for each mouse was normalized to the β-Gal expression in the liver.

**Glucose, insulin, and pyruvate tolerance tests.** After overnight fasting, mice were i.p. injected or treated orally with D-glucose (1–2 g/kg) for glucose-tolerance tests (IGTT or OGTT) or injecting with sodium pyruvate (1–2 g/kg) for pyruvate tolerance tests (PTT). Insulin-tolerance tests (ITT) were performed by i.p. injection of insulin (1 U/kg) 2-h after food removal. The blood glucose content was measured using a glucometer (Roche) before the injection and at 15, 30, 60, and 120 min after agents' injection.

**Total RNA isolation and quantitative PCR analysis.** Total RNA from cells and tissues was extracted by Trizol (Invitrogen) and reverse-transcribed into cDNA using the prime script RT reagent kit with gDNA eraser (Takara). The mRNA levels were determined using the SYBR-green PCR kit (Takara) with ABIPRISM 7900HT Sequence detector (ABI) according to the manufacturer's protocol. The quantitative qPCR primers used in this work are listed in Supplementary Table 1. Ribosomal L32 mRNA levels were used as an internal control.

**Immunoblotting, immunoprecipitation, and immunostaining assays.** Immunoblotting and immunoprecipitation assays were performed as previously described[78], with antibodies recognizing CREB, Ser133-P-CREB, AMPK, P-AMPK, AKT, P-AKT (Cell Signaling), PGC-1α (Merck), CRTC2 (Calbiochem), PCK1 (Santa Cruz), SREBP1 (Santa Cruz), SREBP2 (BD), HA (Convance), LXRα (Santa Cruz), FLAG (Sigma), GST (Covance), HIS (Abmart), GAPDH (AOGMA), α-TUBULIN (Abmart), β-ACTIN (Abmart), and other antibodies (Supplementary Table 3). For immunostaining assays, primary cultured hepatocytes seeded on coverslips were fixed with 4% paraformaldehyde for 20 min, permeabilized with 0.3% Triton X-100 in PBS, and then blocked by 5% normal goat serum and 0.3% Triton X-100 in PBS, followed by incubation with anti-CREB (1:1000), anti-P-CREB (1:1000), or anti-HA (1:3000) antibodies for 1 h at 4 °C. Slides were then washed, and after washing, slides were incubated with fluorochrome-conjugated secondary antibodies (Invitrogen). Slides were then washed and mounted with Vectashield mounting media containing 4, 6-diamidino-2-phenylindole (DAPI).

**ChIP assays.** ChIP was performed according to the fast ChIP protocol[80]. Briefly, hepatocytes were immediately cross-linked in 1% formaldehyde for 10 min, which was then stopped by glycine. The isolated nuclear pellets were sheared by ultrasonic treatment for 50 s around 1 s with 2 s pauses at 50% power output. The sheared chromatin was incubated with normal IgG or an antibody for CREB or CRTC2 for 2 h at 4 °C in the presence of a protease inhibitor. The precipitated chromosome is then pulled down using protein G beads (Sigma) and indicated antibodies, purified by Chelex 100 (Biorad) and quantified by qPCR (ABI 7900). The occupancy of the

transcriptional factor on the CRE sites of indicated genes was normalized to the signal from IgG pulled-down chromatin. Primers used for ChIP-qPCR assays are listed in Supplemental Table 2.

**Measurement of cAMP levels**. At the end of indicated treatments, primary hepatocytes were collected in lysis buffer. The cAMP contents of cell lysates were then determined using cAMP ELISA kits (R&D) and normalized by total protein content as determined by BCA assay (Thermo).

**Methyl thiazolyl tetrazolium assay (MTT)**. Primary hepatocytes were exposed to APC at indicated concentrations for 48-h, then co-incubated with methyl thiazolyl tetrazolium (0.5 mg/mL in PBS, Sigma) solution for 5-h. After removing the solution, the purple crystals were dissolved using DMSO, and the absorbance was measured at 550 nm.

**Glucose production assay**. Primary hepatocytes were exposed to propolis, APC or vehicle control for 1-h prior to 4-h stimulation with glucagon, and then incubated in no-glucose no-phenol-red DMEM (Sigma) supplied with 2 mM sodium pyruvate (Sigma) and 10 mM sodium lactate (Sigma) for 8-h. Glucose levels in the media were detected using a GO kit (Sigma).

**Surface plasmon resonance assay**. BIACORE 100 (GE Healthcare) was employed to perform surface plasmon resonance. Briefly, purified HIS-CREB proteins were captured in the test channel of an NTA-chip (GE, Healthcare) in HEPES buffer (10 mM HEPES, 150 mM NaCl, 3 mM EDTA, pH 7.5). APC solutions with different concentrations were flowed through immobilized HIS-CREB proteins for indicated times, which caused equilibrium response values (RU) to change appropriately. The detected RU values were plotted by time using GraphPad Prism software.

**Isothermal titration calorimetry assay**. Isothermal titration calorimetry was performed using a MicroCal iTC200 calorimeter (Freiburg, Germany). The purified HIS-CREB protein and APC were diluted in buffer (PBS with 500 mM NaCl and 10% glycine, pH 8.0). APC solution was titrated by injection of HIS-CREB. The originally released heat from molecules binding was recorded and analyzed by Origin9 software (Origin Lab) supplied with the instrument. Equilibrium dissociation constant $K_D$ between CREB and APC is indicated by Origen 9.

**Blood chemistry**. The plasma hormone levels of insulin and ghrelin were measured by ELISA kits from Millipore, glucagon and leptin were determined by ELISA kits from R&D.

**Cholesterol assays**. Serum total cholesterol (TC), LDL cholesterol (LDL-C), and HDL cholesterol (HDL-C) levels were detected using commercial kits (Beihua, China), and serum VLDL protein was determined by ELISA (Jiancheng, China).

**TG and NEFA assays**. The plasma TG and NEFA levels were determined using corresponding kits from Beihua (China) and WAKO (Japan), respectively. The TG levels of liver and WAT were tested by TG kit (Applygene, China). The NEFA in the liver and WAT were extracted by chloroform–methanol and then subjected to NEFA assay (WAKO).

**Tissue distribution assay**. Male C57/BL6 mice (18–20 g, $n = 3$ in each group) were randomly sorted into treatment groups. Being fasted for 12-h, mice were treated with oral gavage of the compound APC (20 mg/kg, in DMSO/0.5% HMPC, 5/95, v/v). After 2, 4, and 8-h of administration, the animals were sacrificed, respectively, the indicated tissues, including plasma, liver, kidney, muscle, brain, spleen, pancreas, white and brown fat, were collected and preserved at −80 °C. The APC content in samples was analyzed with a liquid chromatography-mass spectrometry/mass spectrometry (LC–MS/MS) system and normalized by protein content.

**Pharmacokinetic studies**. Male ICR mice (18–20 g, $n = 3$ in each group) were randomly assigned to treatment groups. After 12-h of fasting, baseline blood was collected into a tube containing ethylenediaminetetraacetic acid (EDTA). Animals were then treated orally with compound **A57** or APC (p.o.: 20 mg/kg, in DMSO/0.5% HPMC 5/95, v/v) and intravenous injected the selected compounds (i.v.: 2 mg/kg, in DMSO/EtOH/PEG300/NaCl 5/5/40/50, v/v/v/v). Subsequently, blood samples per mouse were collected at 0.25, 0.5, 1, 2, 4, 8, 24 h for p.o.-treated groups and 0.05, 0.25, 0.75, 2, 4, 6, 8, 24 h for i.v.-treated groups. All samples were centrifuged at 12,000 rpm for 3 min, and the plasma was harvested. Aliquots of plasma samples were stored at −80 °C until analysis. The selected compounds' serum concentrations were determined by liquid chromatography/tandem mass spectrometry (LC–MS–MS). Pharmacokinetic parameters were determined from the selected compounds' serum concentrations by noncompartmental methods using WinNonLin professional version 4.1 (Pharsight Corp., Mountain View, CA).

**Metabolic chamber assay**. Metabolic parameters of mice were determined using Columbus Instruments Comprehensive Lab Animal Monitoring System (CLAMS). The area unit under the curve (AUC) of respiratory exchange ratio (RER) and ambulatory activity (X-TOT) were calculated using Graphpad prism 8.0.

**General information for syntheses and identification of compound A57**. The chemicals were purchased from commercial sources and used without further purification. Analytical thin-layer chromatography (TLC) was performed on 0.15–0.2 mm thickness silica gel plates. $^1$H and $^{13}$C NMR spectra were recorded in dimethyl sulfoxide-$d_6$ (DMSO-$d_6$) on 400 or 500 MHz instruments. Chemical shifts were reported in parts per million (ppm, δ) downfield from tetramethylsilane. Proton coupling patterns were described as singlet (s), doublet (d), triplet (t), quartet (q), quintet (p), doublet of triplets (dt), and multiplet (m). High-resolution mass spectra (HRMS) were measured on a Q-TOF spectrometer. The determination of ee was performed via HPLC analysis. Optical rotations were measured using a 1-mL cell with a 10-mm path length on an Auto pol V polarimeter and are reported as follows: [α] 20D (c: g/100 mL, in solvent). Melting points were measured on a melting point apparatus. The synthetic route of compound **A57** is described in Supplementary Fig. 9a. $^1$H and $^{13}$C NMR spectrum of **A57** as Supplementary Fig. 9b, c. HPLC identification of **A57** can be found in Supplementary Fig. 9d.

**Statistical analysis**. The cell data are presented as mean ± SEM (standard error of the mean) of multiple wells ($n = 3$–6) from one represented experiment. The mouse data are presented as mean ± SEM of indicated biological repeats ($n$ values). The statistical comparisons were performed using unpaired two-tailed $t$ test with Welch's correction between two groups, or one-way ANOVA followed Dunnett's or Tukey's multiple comparisons test among groups more than three compared with a single control, or two-way ANOVA followed Bonferroni's multiple comparisons test among different groups. Each $P$ value is adjusted to account for multiple comparisons, the significance and confidence level is 0.05 (95% confidence interval). Graphpad Prism software (8.0) was used for data statistical analysis. *$P < 0.05$; **$P < 0.01$; ns, $P > 0.05$ not significant.

**Reporting summary**. Further information on research design is available in the Nature Research Reporting Summary linked to this article.

## Data availability
The raw data generated in this study are provided in the Supplementary Information or Source Data file. Source data are provided with this paper.

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

## Acknowledgements

We thank S. Xiang and L.T. Zai for help with protein purification. We also thank M. Wu for technical assistance. Moreover, we thank LetPub (www.letpub.com) for its linguistic assistance during the preparation of this manuscript. This work was supported by grants National Key Research and Development Program (2021YFA08047), and from National Program on Key Basic Research Project of China 973 Program (NBR973, 2012CB524900, 2014CB910500, and 2015CB910304), National Natural Science Foundation of China (NSFC 81390351, 31222028, 81620108027, 21632008, 31200891, 32171160, and 21877118), Chinese Postdoctoral Science Foundation (20100480635), and National Science & Technology Major Program (2013ZX09508104). Strategic Priority Program of Chinese Academy of Sciences and Technology (XDA12040306), Project supported by Shanghai Municipal Science and Technology Major Project, Program of Outstanding Scientific and Technological Innovation Team of Jiangsu Higher Education Institutions, and the Priority Academic Program Development of Jiangsu Higher Education Institutions.

## Author contributions

Y.Q.C., J.W., L.H.H., H.L., and Y.L. planned and designed experiments. Y.Q.C., W.P.F., Z.Z., R.W., Q.X., Y.X.L., L.Q.Z., and L.W. carried out the experiments in vitro and in vivo. H.Y.Y. and L.H.H. executed chemical work including propolis compounds isolation and synthesis APC. J.W., Y.B.W., and H.L. performed compound structure modification and synthesis A57. Y.Q.C, J.W., L.H.H., H.L., and Y.L. prepared the manuscript.

## Competing interests

The authors declare no competing interests.
