## [Peer Review File · Nature Communications]

Reviewers' Comments:

Reviewer #1:

Remarks to the Author:

In this manuscript, the authors showed the efficacy of the artemillin C in disrupting CREB/CRTC2 interaction, which could explain the improved metabolic phenotypes in DIO mice and db/db mice. In vitro study showed the critical effect of this drug on blocking CREB/CRTC2-driven hepatic gluconeogenesis, as well as hepatic lipid biosynthesis by reducing expression of LXRA and SREBP2. Furthermore, the authors showed the efficacy of this drug in improving metabolic homeostasis in obese mouse models, suggesting a potential therapeutic application of their findings. Although the molecular mechanism by which APC and its derivatives specifically target CRTC2 is clear, major concerns regarding the efficacy of this reagent on whole body physiology should be further addressed.

1. Since CRTC2 is expressed ubiquitously, the authors should verify whether APC can specifically affect hepatic CRTC2 to modulate glucose and lipid metabolism in a tissue-specific manner (Please refer to the specific comments below).
2. A recent publication showed that CRTC2 inhibits hepatic lipogenesis by controlling SREBP-1c processing in the liver (Nature. 2015 Aug 13;524(7564):243-6. doi: 10.1038/nature14557. Epub 2015 Jul 6), which is not compatible with the current study. Although, the authors showed the effect of APC on SREBP-1/2 expression by using WT and CRTC2 null hepatocytes, in vivo studies should be also performed by using WT and CRTC2 KO mice.
3. A recent study showed a role for CRTC2 in regulating SREBP2 expression at the transcription level, and CRTC2 null mice thus displayed reduced hepatic cholesterol synthesis (Hepatology. 2017 Aug;66(2):481-497. doi: 10.1002/hep.29206. Epub 2017 Jun 27. Erratum in: Hepatology. 2018 Mar;67(3):1188.) The authors cited this paper, but only indicated that this work only showed the transcriptional mechanism for CRTC2 in regulating SREBP2 expression. Rather, the study shown in Hepatology fully addressed in vivo physiology by using CRTC2 null mice, unlike the current study that did not utilize the CRTC2 null mice.
4. Another recent publication using CRTC2 liver-specific KO mice were published by a different group recently, showing that depletion of CRTC2 improved fatty liver and obese phenotypes in DIO mice by activation of PPAR alpha-FGF21 axis (Nat Commun. 2017 Nov 30;8(1):1878. doi: 10.1038/s41467-017-01878-6.). The authors should explore whether the pathways shown in CRTC2 LKO mice could be also detected by APC treatment in the liver.
5. As mentioned previously, CRTC2 is expressed ubiquitously, and plays an important role in various metabolic tissues such as adipose tissues, pancreatic beta cells, and intestinal L cells (Cell Rep. 2018 Sep 18;24(12):3180-3193. doi: 10.1016/j.celrep.2018.08.055.; FASEB J. 2018 Mar;32(3):1566-1578. doi: 10.1096/fj.201700845R. Epub 2018 Jan 3.; Cell Rep. 2015 Feb 24;10(7):1149-57. doi: 10.1016/j.celrep.2015.01.046. Epub 2015 Feb 19.). The authors should confirm/deny whether the in vivo effect of APC could be potentially associated with changes in CRTC2 activity in these tissues.
6. As a minor point, the authors should utilize ANOVA test in comparing values among groups that are more than 2, even if one only compares two values. Thus, the authors should repeat the statistical analysis by using ANOVA in most cases.

Reviewer #2:

Remarks to the Author:

The manuscript NCOMMS-18-34605 from Prof Liu and colleagues, entitled "A Propolis-Derived Small Molecule Ameliorates Metabolic Syndrome in Obese Mice via Targeting CREB/CRTC2 Transcriptional Complex" reports the discovery that the small molecule artemillin C, from Propolis, disrupts CRTC-CREB-mediated transcription. The authors further show that artemillin C treatment blocks CRTC-CREB-mediated transcription of gluconeogenic genes and LXRA. Treatment of DIO and db/db mice with artemillin C was associated with lowering of body weight, fasting blood glucose, plasma lipids, cholesterol, and liver steatosis. The authors link these in vivo effects to artemillin C blockade of CRTC2-CREB transcription of gluconeogenic genes and LXR alpha in liver and hepatocytes. Lastly, the authors employ synthetic chemistry to generate a more potent

inhibitor A57 that shows similar effects in vivo on gluconeogenic gene expression and fasting blood glucose lowering. These original results and conclusions are well-supported by a broad range of molecular, biochemical, cell-based and in vivo experiments. Additional control experiments were provided to support the author's interpretation of data. The work as a whole is of broad interest for the metabolic and transcriptional biology research areas. Below are my review comments and suggested experiments to improve the manuscript. Please note, that I did not provide a critical review of the chemical synthesis of A57, since this is beyond the scope of my scientific expertise.

Important considerations:

a) The specificity of propolis-derived small molecules for disrupting CRTC2-CREB interaction could be better supported by additional experiments.

- While the authors show in pull-down assays that artepillin C does not impair P300-CREB interaction, a more sensitive assay could be used. Does propolis, artepillin C, or A57-derivatives affect P300-CREB interactions, KID-KIX interactions or CBP/P300 histone acetyltransferase activity that facilitate CREB-mediated transcription?

- Mechanistically, does artepillin C or A57 compete with CRTC binding of the B-ZIP domain of CREB?

b) Artepillin C exhibits low potency. Including a derivative that shows no impact in the screen from 1F as a negative control for in vitro pharmacology (Figure 2F) and binding experiments (ITC, SPR; Figures 2E and F) would help alleviate concerns regarding non-specific effects due to micromolar concentrations of compound in the assays.

c) Given the high μM IC_{50} 's for Artepillin C and A57, it is unclear whether these concentrations of compound are reached in vivo in blood and tissue and for how long. Data for plasma and tissue exposure over time to support the choice of dose levels and frequency of administration for in vivo experiments is necessary.

d) The in vivo artepillin C treatment data provide compelling evidence showing improvements in body weight and whole-body metabolism, that could be mediated through CRTC2-CREB actions in multiple tissues or via other mechanisms.

- Are glucagon levels altered with artepillin C treatment in DIO or db/db mice?

- Is artepillin C treatment capable of blocking constitutively nuclear CRTC2 S171A-mediated upregulation of gluconeogenic (and LXR α) gene transcription or luciferase reporter in vivo in liver and/or in hepatocytes?

- Do body weight and/or decreases in adiposity precede blood glucose lowering during the 5 weeks of artepillin C treatment?

- Additional discussion comparing the results in this manuscript with findings from liver specific CRTC2 knockout mice would be helpful and reference to this relevant paper should be included (PMID:29192248)

e) Please use appropriate statistical analyses for comparisons between 3 or more groups. There are instances where a two-way ANOVA and two-way ANOVA with repeated measures should also be used in the manuscript.

Other considerations:

a) It is unclear if the improvements in glucose and insulin tolerance tests are mainly driven by the lower glucose levels at T0. The AUC for these tolerance tests should be provided with appropriate statistical analyses.

b) Effects on daily food intake with APC treatment is provided for db/db animals. If possible, please the effects of APC on daily food intake in DIO mice would provide consistency.

c) For body composition analyses, the absolute values for fat and lean mass would be helpful to ensure the decrease in the fat/lean ratio is driven by reductions in adiposity. Insulin and leptin levels are shown for db/db experiment. Please also show leptin levels for db/db experiment. It is

not clear why leptin levels are not reduced if adiposity is reduced with artemillin c. Please discuss.

e) Does APC treatment also reduce liver steatosis in db/db mice or are these effects only observed in DIO mice?

f) Figure 3 D is lacking the minus or plus signs to indicate glucagon stimulation

g) Figure 3A: consider adding space between the symbol legends and the titles for each graphs

Reviewer #3:

Remarks to the Author:

In the present article entitled: A Propolis-Derived Small Molecule Ameliorates Metabolic Syndrome in Obese Mice via Targeting CREB/CRTC2 Transcriptional Complex, the authors focused to investigate the effect of artemillin C (APC), a small-molecular fraction isolated from Brazilian green propolis, on the disruption of CREB-CRTC2 interaction which, in turn, acts as a key regulator of hepatic gluconeogenesis. The data presented in this manuscript is very interesting and important in this field, but several points (Major corrections) should be seriously taken in consideration for the following reasons:

1- The authors are advised to revise the manuscript against English language errors that are present throughout the manuscript.

2- The authors must define the abbreviations at their first use because several abbreviations were not defined throughout the manuscript.

3- In the last paragraph of the introduction section, the authors stated that "Here, we reported that CREB/CRTC2 was the molecular target for Brazilian green propolis ... to the end of this section". This paragraph should be deleted and the authors should add the hypothesis and aims of the present work.

4- In the results section, lines 106-108 and lines 111-112, these sentences must be removed from the results section and added to the discussion section. Additionally, line 209 the following abbreviations were incorrectly defined "LDL cholesterol (LDLC), HDL cholesterol (HDLC).

5- In the materials and methods section, lines 530-536 the authors must add the information and concentrations of the antibodies that were used in this section.

6- The most critical point in this work is: in figure 2, figure 3D, figure 5B &F, figure 6B and figure 7C, the authors demonstrated the protein bands of one representative experiment. These experiment must be repeated for at least three times and the phosphorylated proteins must be normalized to the total relevant proteins, as well as, the expression of other proteins must be normalized to the total loading proteins (GAPDH or beta-actin). Then, accumulated data must be presented as bar figure (normalized values +/-SD) under each protein bands.

Gamal Badr

<https://www.scopus.com/authid/detail.uri?authorId=35338908600>

http://www.aun.edu.eg/membercv.php?M_ID=393

Reviewer #4:

Remarks to the Author:

In this contribution, Ya-qiong Chen et al. described an artemillin's novel analog that exhibits higher inhibitory activity on CREB-CRTC2 interaction.

The medicinal chemistry part is incomplete which seriously affects the quality of the study.

The design of compound A57 is not explained although the authors indicate that they made a series of molecules from artemillin as a lead compound!!

The entire series must be included in the manuscript to fully understand the strategy that led to compound A57. With all series, the effect of substitutions at the 3-position of the phenyl and the effect of the presence of the amide moiety instead of the carboxylic acid can be elucidated

How did the authors arrive with this chiral compound? a racemic mixture will be active or not??

All intermediates must be characterized (^1H , ^{13}C , HMRS).

The authors indicate that the goal was to develop ``small molecules with satisfactory drug-available``. I find no comparison with artemisinin. A57 (ClogP: 6.2) is more lipophilic than artemisinin (ClogP: 5.4). This difference in activity is not due to an increase of the lipophilicity?

A comparison of the pharmacokinetic properties is necessary.

Reviewer #1 (Remarks to the Author):

In this manuscript, the authors showed the efficacy of the artemillin C (APC) in disrupting CREB/CRTC2 interaction, which could explain the improved metabolic phenotypes in DIO mice and *db/db* mice. In vitro study showed the critical effect of this drug on blocking CREB/CRTC2-driven hepatic gluconeogenesis, as well as hepatic lipid biosynthesis by reducing expression of LXRA and SREBP2. Furthermore, the authors showed the efficacy of this drug in improving metabolic homeostasis in obese mouse models, suggesting a potential therapeutic application of their findings. Although the molecular mechanism by which APC and its derivatives specifically target CRTC2 is clear, major concerns regarding the efficacy of this reagent on whole body physiology should be further addressed.

- 1. Since CRTC2 is expressed ubiquitously, the authors should verify whether APC can specifically affect hepatic CRTC2 to modulate glucose and lipid metabolism in a tissue-specific manner (Please refer to the specific comments below).**

Response: We agree with Reviewer 1's concern. In order to determine the effect of artemillin C (APC) in a tissue-specific manner, we measured the concentration of APC in different tissues, including the plasma, liver, muscle, white and brown adipose tissues, spleen, brain, kidney and the pancreas. Two to 8 hours after APC oral administration, APC tissue distribution analysis revealed that the highest relative content of APC (about 916.8 ± 207.3 ng/g, 3.05 ± 0.69 μ M) was in liver 4 h after *p.o.* (Fig. 4K). Although the relative CRTC2 protein level in the liver was not the highest of the tissues/organs we checked (Suppl. Fig 4H), the normalized level of APC content relative to the CRTC2 protein level in each tissue/organ was the highest in the liver among these tissues/organs (Fig. 4L). These results indicated that the liver is the major target tissue of APC *in vivo* and suggested that APC modulated glucose and lipid metabolism in a liver-specific manner.

- 2. A recent publication showed that CRTC2 inhibits hepatic lipogenesis by controlling SREBP-1c processing in the liver (Nature. 2015 Aug 13;524(7564):243-6. doi: 10.1038/nature14557. Epub 2015 Jul 6), which is not compatible with the current study. Although, the authors showed the effect of APC on SREBP-1/2 expression by using WT and CRTC2 null hepatocytes, in vivo studies should be also performed by using WT and CRTC2 KO mice. (Nature. 2015 Aug 13;524(7564):243-6. doi: 10.1038/- The CREB coactivator CRTC2 controls hepatic lipid metabolism by regulating SREBP1.**

Response: We thank Reviewer 1 for his/her comments; however, we do not agree with Reviewer 1's opinion that our results are not compatible with the study in the aforementioned *Nature* paper. In that work, CRTC2 was reported to inhibit the SREBP-1c protein mature process in the cytosol *via* interaction with SEC31A, a

subunit of COPII, attenuating COPII-dependent pre-SREBP1 vesicle trafficking from the ER to the Golgi. We checked whether APC affected the affinity between CRTC2 and SEC31A. Our co-immunoprecipitation results indicated that APC had no impact on the association between CRTC2 and SEC31A (Suppl. Fig. 5B), which excluded the possibility that APC regulates CRTC2-dependent-pre SREBP1 trafficking and maturation. This result also suggests that APC regulates the transcriptional function of nuclear CRTC2 but not its cytosolic function.

According to Reviewer 1's suggestion, we did check the effect of APC on SREBP-1/2 expression in WT and CRTC2-KO mice. We found that APC did not further reduce the protein levels of SREBP1 and 2 in the livers of CRTC2-KO mice (Fig. 5G), which suggested that the capability of APC to decrease the transcription of the master factor SREBPs required for the transcriptional activity of CREB/CRTC2. Therefore, the effects of APC *in vivo* were blocked by CRTC2-KO.

- 3. A recent study showed a role for CRTC2 in regulating SREBP2 expression at the transcription level, and CRTC2 null mice thus displayed reduced hepatic cholesterol synthesis (Hepatology. 2017 Aug;66(2):481-497. doi: 10.1002/hep.29206. Epub 2017 Jun 27. Erratum in: Hepatology. 2018 Mar;67(3):1188.) The authors cited this paper, but only indicated that this work only showed the transcriptional mechanism for CRTC2 in regulating SREBP2 expression. Rather, the study shown in Hepatology fully addressed *in vivo* physiology by using CRTC2 null mice, unlike the current study that did not utilize the CRTC2 null mice.**

Response: According to Reviewer 1's suggestion, we checked the effects of APC on cholesterol, FFA and TG levels in CRTC2-KO mice. Compared with the decrease in the levels of plasma and liver cholesterol, as well as liver TG by APC treatment in wild type mice, these levels were little affected by APC treatment in CRTC2-KO mice (Fig. 5H; Suppl. Fig. 5C, 5D), implying the necessity of CRTC2 for the beneficial effects of APC on lipid metabolism *in vivo*. Again, APC reduction of SREBP2-mediated lipid synthesis was blocked by CRTC2-KO *in vivo*.

- 4. Another recent publication using CRTC2 liver-specific KO mice were published by a different group recently, showing that depletion of CRTC2 improved fatty liver and obese phenotypes in DIO mice by activation of PPAR alpha-FGF21 axis (Nat Commun. 2017 Nov 30; 8(1): 1878. doi: 10.1038/s41467-017-01878-6.). The authors should explore whether the pathways shown in CRTC2 LKO mice could be also detected by APC treatment in the liver.**

Response: Based on Reviewer 1's suggestion, we checked the effects of APC on FGF21. We found that APC and A57 increased the plasma and hepatic levels of

FGF21 in WT mice, whereas, these did not further increase them in CRT2-KO mice (Suppl. Fig. 10F and 10G), suggesting that APC and A57 may improve energy metabolism through multiple CREB/CRT2-mediated pathways, including the hepatic PPAR α -FGF21 axis mentioned in this work.

- 5. As mentioned previously, CRT2 is expressed ubiquitously, and plays an important role in various metabolic tissues such as adipose tissues, pancreatic beta cells, and intestinal L cells (Cell Rep. 2018 Sep 18;24(12):3180-3193. doi: 10.1016/j.celrep.2018.08.055.; FASEB J. 2018 Mar;32(3):1566-1578. doi: 10.1096/fj.201700845R. Epub 2018 Jan 3.; Cell Rep. 2015 Feb 24;10(7):1149-57. doi: 10.1016/j.celrep.2015.01.046. Epub 2015 Feb 19.). The authors should confirm/deny whether the *in vivo* effect of APC could be potentially associated with changes in CRT2 activity in these tissues.**

Response: To address Reviewer 1's suggestion, we checked the bioavailability of APC in different tissues. As shown in the response to Question 1, we found that the liver was the major target tissue of APC activity *in vivo* (Fig. 4K and 4L). The liver-specific distribution of APC reduced the possible side effects in other tissues, such as those mentioned by Reviewer 1.

- 6. As a minor point, the authors should utilize ANOVA test in comparing values among groups that are more than 2, even if one only compares two values. Thus, the authors should repeat the statistical analysis by using ANOVA in most cases.**

Response: We thank Reviewer 1 for the suggestion and have used an ANOVA to retest the statistical analysis as indicated in the figure legends.

Reviewer #2 (Remarks to the Author):

The manuscript NCOMMS-18-34605 from Prof Liu and colleagues, entitled "A Propolis-Derived Small Molecule Ameliorates Metabolic Syndrome in Obese Mice via Targeting CREB/CRTC2 Transcriptional Complex" reports the discovery that the small molecule artemillin C (APC), from Propolis, disrupts CRTC-CREB-mediated transcription. The authors further show that artemillin C treatment blocks CRTC-CREB-mediated transcription of gluconeogenic genes and LXRalpha. Treatment of DIO and db/db mice with artemillin C was associated with lowering of body weight, fasting blood glucose, plasma lipids, cholesterol, and liver steatosis. The authors link these in vivo effects to artemillin C blockade of CRTC2-CREB transcription of gluconeogenic genes and LXR alpha in liver and hepatocytes. Lastly, the authors employ synthetic chemistry to generate a more potent inhibitor **A57** that shows similar effects in vivo on gluconeogenic gene expression and fasting blood glucose lowering. These original results and conclusions are well-supported by a broad range of molecular, biochemical, cell-based and in vivo experiments. Additional control experiments were provided to support the author's interpretation of data.

The work as a whole is of broad interest for the metabolic and transcriptional biology research areas. Below are my review comments and suggested experiments to improve the manuscript. Please note, that I did not provide a critical review of the chemical synthesis of **A57**, since this is beyond the scope of my scientific expertise.

Important considerations:

a) **The specificity of propolis-derived small molecules for disrupting CRTC2-CREB interaction could be better supported by additional experiments. While the authors show in pull-down assays that for APC does not impair P300-CREB interaction, a more sensitive assay could be used. Does propolis, APC, or A57-derivatives affect P300-CREB interactions, KID-KIX interactions or CBP/P300 histone acetyltransferase activity that facilitate CREB-mediated transcription? mechanistically, does APC or A57 compete with CRTC binding of the B-ZIP domain of CREB?**

Response: As Reviewer 2 suggested, we checked the effect of APC and **A57** on the interaction between CREB-KID and CBP-KIX using a two-hybrid assay as this is more sensitive than co-immunoprecipitation. The results showed that neither APC nor **A57** reduced *Gal4*-Luc reporter activity (Suppl. Fig. 2H), suggesting that APC and **A57** did not affect CREB-CBP association. Similar to the results of our co-immunoprecipitation data (Fig. 2C), APC had little effect on the interaction between overexpressed P300 and GST-CREB (Suppl. Fig. 2A). However, the histone acetyltransferase activity of CBP, which is critical for transcription at CRE sites, was significantly inhibited by APC in a dose dependent manner (Fig. 2H). Moreover, APC dose-dependently inhibited FLAG-CRTC2 binding with MYC-CREB-ZIP as shown by co-immunoprecipitation (Fig. 2G), implying a competitive role for APC in the association between CRTC2 and the CREB-

ZIP domain.

b) APC exhibits low potency. Including a derivative that shows no impact in the screen from 1F as a negative control for in vitro pharmacology (Figure 2F) and binding experiments (ITC, SPR; Figures 2E and F) would help alleviate concerns regarding non-specific effects due to micromolar concentrations of compound in the assays.

Response: As Reviewer 2 suggested, we chose P4 (*p*-coumaric acid ethyl ester, Fig. 1F) as a negative control compound and used it our ITC assays. As shown in Suppl. Fig. 2G, P4 did not interact with HIS-CREB, which excluded a nonspecific interaction between popolis small molecular compounds and CREB protein.

c) Given the high μM IC₅₀'s for APC and A57, it is unclear whether these concentrations of compound are reached in vivo in blood and tissue and for how long. Data for plasma and tissue exposure over time to support the choice of dose levels and frequency of administration for in vivo experiments is necessary.

Response: As Reviewer 2 suggested, we checked the distribution of oral APC (20 mg/kg) *in vivo*. The highest content of APC was 916.8 ± 207.3 ng/g (3.06 ± 0.69 μM) as detected in the liver at 4 h after *p.o.* (Fig. 4K), which was comparable to the micromolar concentration (5–10 μM) used in our *in vitro* assays. To determine the hypoglycemic effect of APC *in vivo*, we *i.p.* administered APC to DIO obese mice daily at different doses (0, 10, and 20 mg/kg) for 5 weeks. The results demonstrated that 10 mg/kg APC decreased fat mass, the fat/lean ratio, as well as reducing the levels of plasma cholesterol and TG (Fig. 4B–E). Moreover, 20 mg/kg APC further reduced levels of plasma cholesterol as well as liver NEFA and glycerol (Fig. 4E, 4G and 4I). Additionally, the insulin sensitivity of DIO mice was remarkable increased by 20 mg/kg APC (Fig. 7F–H). In other words, 20 mg/kg APC was effective in model mice but did not lead to hypoglycemia in lean mice (Suppl. Fig. 8A and 8B). Therefore, we selected 20 mg/kg APC for follow-up work *in vivo*.

We further investigated the pharmacokinetic properties of A57 and APC administered both orally (*p.o.*) and intravenously (*i.v.*) in ICR mice. The results (Table 1) showed that plasma was exposed to a maximum concentration of APC of 510 ng/mL (C_{max} APC *p.o.*) at 0.25 h (T_{max} APC *p.o.*), with a peak concentration of A57 of 165 ng/mL (C_{max} A57 *p.o.*) at 0.67 h (T_{max} A57 *p.o.*). Moreover, the half-life of *p.o.* APC and A57 were 2.11 h ($T_{1/2}$ APC, *p.o.*) and 3.29 h ($T_{1/2}$ A57, *p.o.*), respectively. Although the half-life of APC and A57 were both shorter than 4 h, onetime daily administration could effectively reduce fasting blood glucose, so we used this regime in this work.

Pharmacokinetic assessment suggested that plasma exposure maximum concentrations of APC were relatively lower than the IC₅₀ determined by two-hybrid assays. However,

the compound enrichment in target tissue liver could help to maintain stable and sustained concentration of APC in the liver, which brings benefits for tissue-specific regulation of glucose and lipid metabolism in the liver. Therefore, the choice of dose and frequency of administration *in vivo* was dependent on our first experiments in mice (Figs. 4–5).

d) The *in vivo* APC treatment data provide compelling evidence showing improvements in body weight and whole-body metabolism, that could be mediated through CRTC2-CREB actions in multiple tissues or via other mechanisms.

Response: We agree with Reviewer 2's comment that APC's effects could be mediated through CRTC2-CREB actions in multiple tissues or *via* other mechanisms. However, APC tissue-distribution analysis showed that the liver was the main target of APC activity (Fig. 4K–L). Moreover, we also found that APC might improve lipid metabolism by enhanced FGF21 expression through inhibiting CRTC2 transcription (Suppl. Fig. 10F and 10G). Therefore, APC improved glucose/lipid metabolism mainly through inhibiting hepatic CREB/CRTC2.

- Are glucagon levels altered with APC treatment in DIO or *db/db* mice?

Response: To address Reviewer 2's question, we checked the plasma glucagon levels in mice treated with APC, and did not detect any effects of APC on glucagon levels in DIO or *db/db* mice (Suppl. Fig. 7E and F).

- Is APC treatment capable of blocking constitutively nuclear CRTC2 S171A-mediated upregulation of gluconeogenic (and LXR α) gene transcription or luciferase reporter *in vivo* in liver and/or in hepatocytes?

Response: As implied by Reviewer 2's question, we checked the capability of APC to inhibit CRTC2-S171A-mediated upregulation of gluconeogenic and LXR α gene transcription in mice. A luciferase reporter assay showed that APC decreased the *G6p-luc* reporter activity induced by overexpression of CRTC2-S171A in primary hepatocytes (Fig. 3E). Correspondingly, APC also decreased the mRNA levels of the gluconeogenic genes *G6p*, *Pck1* and *Pgc1 α* (Fig. 3E and 3F), as well as *Lxr α* (Fig. 6F), which were stimulated by the overexpression of CRTC2-S171A.

- Do body weight and/or decreases in adiposity precede blood glucose lowering during the 5 weeks of APC treatment?

Response: Our data indicated that oral administration of APC or A57 remarkably

decreased blood glucose levels after 3 days, but no significant changes in body weight or adiposity were detected in these animals at this time.

The decreasing of body weight needed at least 3 weeks of administration to detect (Fig. 4A, Suppl. Fig. 4B).

- Additional discussion comparing the results in this manuscript with findings from liver specific CRTC2 knockout mice would be helpful and reference to this relevant paper should be included (PMID:29192248)

Response: We referred and discussed (page 26, paragraph 2, the last 3 lines, citation 24) the paper mentioned by Reviewer 2.

e) Please use appropriate statistical analyses for comparisons between 3 or more groups. There are instances where a two-way ANOVA and two-way ANOVA with repeated measures should also be used in the manuscript.

Response: We thank Reviewer 2 for the suggestion and have used the corresponding ANOVA tests to perform statistical analysis as indicated in the figure legends.

Other considerations:

a) It is unclear if the improvements in glucose and insulin tolerance tests are mainly driven by the lower glucose levels at T0. The AUC for these tolerance tests should be provided with appropriate statistical analyses.

Response: We thank Reviewer 2 for the suggestion and have added AUCs analyses for these tolerance tests (Fig. 1A, Fig. 7F–H, and Suppl. Fig. 7J–L). According to inhibition of gluconeogenesis, lower blood glucose at T0 by treatment of APC and A57, at least partially, contributed to improvement in insulin sensitivity.

b) Effects on daily food intake with APC treatment is provided for db/db animals. If possible, please the effects of APC on daily food intake in DIO mice would provide consistency.

Response: We thank Reviewer 2 for the suggestion and have added daily food intake in DIO mice treated with APC (Suppl. Fig. 7H).

c) For body composition analyses, the absolute values for fat and lean mass would be helpful to ensure the decrease in the fat/lean ratio is driven by reductions in adiposity. Insulin and leptin levels are shown for db/db experiment. Please also

show leptin levels for db/db experiment. It is not clear why leptin levels are not reduced if adiposity is reduced with APC. Please discuss.

Response: We thank Reviewer 2 for this suggestion and have added the absolute values for fat and lean mass levels in DIO and *db/db* mice (Fig. 4B left and Suppl. Fig. 4C left). We also included the leptin levels for *db/db* mice experiments (Fig. 7L) and discuss this in the manuscript (page 26, paragraph 2, the last sentence) as follows: “APC decrease of body weight and plasma leptin levels in *db/db* mice was associated with increased insulin sensitivity and respiratory exchange ratio (RER, Fig.7), which yields benefits for improving whole body metabolism.”

e) Does APC treatment also reduce liver steatosis in *db/db* mice or are these effects only observed in DIO mice?

Response: We thank Reviewer 2 for the suggestion and checked the effect of APC on liver steatosis in *db/db* mice. H&E stain showed that APC reduced lipid droplets in the livers of *db/db* mice (Suppl. Fig. 4G), implying the alleviation of liver steatosis in these animals by APC.

f) Figure 3 D is lacking the minus or plus signs to indicate glucagon stimulation

Response: We thank Reviewer 2 for the suggestion, and have corrected this in Fig. 3D.

g) Figure 3A: consider adding space between the symbol legends and the titles for each graph

Response: We thank Reviewer 2 for the suggestion and have corrected this in Fig. 3A.

Reviewer #3 (Remarks to the Author):

In the present article entitled: A Propolis-Derived Small Molecule Ameliorates Metabolic Syndrome in Obese Mice via Targeting CREB/CRTC2 Transcriptional Complex, the authors focused to investigate the effect of artipillin C (APC), a small-molecular fraction isolated from Brazilian green propolis, on the disruption of CREB-CRTC2 interaction which, in turn, acts as a key regulator of hepatic gluconeogenesis. The data presented in this manuscript is very interesting and important in this field, but several points (Major corrections) should be seriously taken in consideration for the following reasons:

- 1- The authors are advised to revise the manuscript against English language errors that are present throughout the manuscript.**

Response: We thank Reviewer 3 for this suggestion and have had this manuscript revised by professional English editors. A certificate of editing is available upon request.

- 2- The authors must define the abbreviations at their first use because several abbreviations were not defined throughout the manuscript.**

Response: We thank Reviewer 3 for the suggestion and have added abbreviations for each technical term at its first appearing in the manuscript.

- 3- In the last paragraph of the introduction section, the authors stated that “Here, we reported that CREB/CRTC2 was the molecular target for Brazilian green propolis ... to the end of this section”. This paragraph should be deleted and the authors should add the hypothesis and aims of the present work.**

Response: We thank Reviewer 3 for the suggestion and have revised the end of introduction to add the hypothesis and aims of the present work (page 5, the last two sentences). This now reads as follows “In our preliminary study, we identified some natural inhibitors of CREB/CRTC2 in propolis extracts. To characterize these native CREB/CRTC2 inhibitors, we then established a platform to select, identify and confirm these novel inhibitors targeting CREB/CRTC2 interaction, which should be suitable candidates for developing novel anti-diabetic drugs.”

- 4- In the results section, lines 106-108 and lines 111-112, these sentences must be removed from the results section and added to the discussion section. Additionally, line 209 the following abbreviations were incorrectly defined “LDL cholesterol (LDLC), HDL cholesterol (HDLC).**

Response: We thank Reviewer 3 for the suggestion and moved the sentence in lines 106–108 and lines 111–112 to the Discussion section (page 24, paragraph 2 ,first sentence, and page 25, paragraph 1, the last 1 sentence), and changed the abbreviation of “LDL cholesterol (LDLC)” to “LDL cholesterol (LDL-C)” and “HDL cholesterol (HDLC)” to “HDL cholesterol (HDL-C)” (page 12, paragraph 1, line 7-8).

- 5- In the materials and methods section, lines 530-536 the authors must add the information and concentrations of the antibodies that were used in this section.**

Response: We thank Reviewer 3 for the suggestion and had added the information and concentrations of the antibodies that were used in this section in Supplementary table 3.

- 6- The most critical point in this work is: in figure 2, figure 3D, figure 5B &F, figure 6B and figure 7C, the authors demonstrated the protein bands of one representative experiment. These experiments must be repeated for at least three times and the phosphorylated proteins must be normalized to the total relevant proteins, as well as, the expression of other proteins must be normalized to the total loading proteins (GAPDH or beta-actin). Then, accumulated data must be presented as bar figure (normalized values +/-SD) under each protein bands.**

Response: We thank Reviewer 3 for this suggestion. We actually repeated these experiments three times and have just shown one representative blot in the manuscript for brevity’s sake. We included corresponding information in the legends of each figures mentioned above. We performed the densitometry analysis with this data as suggested by Reviewer 3, have added the results as bar graphs in these figures and provided corresponding information in the figure legends.

Reviewer #4

In this contribution, Ya-qiong Chen et al. described an artemisinin's novel analog that exhibits higher inhibitory activity on CREB-CRTC2 interaction.

Question 1: The medicinal chemistry part is incomplete which seriously affects the quality of the study.

Response: We are grateful for the reviewer's suggestions in this regard. We have revised the medicinal chemistry sections in our revised manuscript. We added more information on the design of compound **A57**, the SAR analysis of this series of compounds, the racemic and enantiomer of compound **A57**, as well as adding the different inhibitory activities and a pharmacokinetic profiles comparison between **A57** and APC, etc. We hope that this additional information addresses the reviewer's concerns about the medicinal chemistry information.

Question 2: The design of compound A57 is not explained although the authors indicate that they made a series of molecules from artemisinin as a lead compound!!

Response: To address the reviewer's suggestions, we have added the design of compound **A57** to our revised manuscript. We revised this part as follow: "Initially, we reserved the pharmacophore α,β -unsaturated ketone, and then assessed various substitutions on this pharmacophore in order to increase the inhibitory activity of CREB/CRTC2 protein-protein interactions with APC. A series of novel α,β -unsaturated ketone derivatives were then designed and synthesized (Fig. 8A). Among them, compound **A57** showed better capability for inhibiting CREB/CRTC2 than APC (IC_{50} 0.74 μ M vs. 24.5 μ M, respectively, Fig. 8B)."

Question 3: The entire series must be included in the manuscript to fully understand the strategy that led to compound A57. With all series, the effect of substitutions at the 3-position of the phenyl and the effect of the presence of the amide moiety instead of the carboxylic acid can be elucidated

Response: We are grateful to the reviewer's suggestions and comments. We have added the SAR analysis of this series compounds in our revised manuscript. We revised this part in discussion as follow: "All the synthesized compounds were evaluated *in vitro* for inhibition of CREB/CRTC2. Initially, our SAR started from APC. We then reserved the pharmacophore α,β -unsaturated acid, and assessed the effect of substitutions at the 3-position of the phenyl. The mono-substituted compounds demonstrated a regiochemical preference of para > ortho > meta. Both electron-donating and electron-withdrawing groups on the phenyl ring were tolerated for CREB/CRTC2 inhibition. Among them, phenyl substitution exhibited good inhibitory activity against CREB/CRTC2 interaction. Then, an R₁ substitution was explored. Amide compounds displayed better inhibitory activity than carboxylic acid derivatives."

Question 4: How did the authors arrive with this chiral compound? a racemic mixture will be active or not??

Response: In response to the reviewer’s question, a racemic mixture and another enantiomer were synthesized and we determined their inhibitory activities *in vitro*. The results are described as follows in the revised manuscript (page 20, paragraph 1): “Considering **A57** was a chiral compound, a racemic mixture and another enantiomer were synthesized and we then determined their inhibitory activities *in vitro*. The results revealed that inhibitory activity of **A57** (*R*-enantiomer) was higher than **A1101** (racemic) or **A58** (*S*-enantiomer) (Fig. 8C)”.

Question 5: All intermediates must be characterized (1H, 13C, HMRS).

Response: We have added the characterizations (¹H, ¹³C, HMRS) of all intermediates in our revised manuscript (suppl. Fig. 9A).

Question 6: The authors indicate that the goal was to develop “small molecules with satisfactory drug-available”. I find no comparison with artemisinin. A57 (ClogP: 6.2) is more lipophilic than artemisinin (ClogP: 5.4). This difference in activity is not due to an increase of the lipophilicity?

Response: We are very grateful for the reviewer’s suggestions and comments. The difference in activity was due to an increase in the lipophilicity. We added a comparison of lipophilicity of compound **A57** with APC in our revised manuscript. The pertinent section now reads as follows: “Additionally, **A57** (cLog *P* = 6.2) was more potent than APC (cLog *P* = 5.4), due to increasing the lipophilicity, and this resulted in higher activity for disrupting CREB/CRTC2 interaction.”

Question 7: A comparison of the pharmacokinetic properties is necessary.

Response: We thank the reviewer for this suggestion. We have added a comparison of the pharmacokinetic properties of compounds **A57** and APC. The details are as follow: “We further investigated the bioavailability of **A57** and APC administered orally (*p.o.*) or intravenously (*i.v.*) in ICR mice. The results (Table 1) showed that compound **A57** displayed a good AUC (648 ng·h/mL) and better oral bioavailability (25.3% vs 14.4%) compared with APC. This data indicated that the increased effective bioactivity of **A57** was tightly associated with increased bioavailability. Collectively, the above data suggested that this series of compounds could serve as possible lead compounds for the development of CREB/CRTC2 inhibitors”.

Table 1. Pharmacokinetic Properties of Compounds **A57** and APC in ICR mice

compd.	admin.	dose (mg/kg)	Cmax (ng/mL)	Tmax (h)	T1/2 (h)	AUC0-t (ng·h/mL)	MRT (h)	Cl (L/h/kg)	Fpo (%)
A57	p.o.	20	165	0.67	2.11	648	3.43	-	25.3
	i.v.	2	-	-	1.21	256	1.40	131	-
APC	p.o.	20	510	0.25	3.29	616	4.33	-	14.4
	i.v.	2	-	-	5.89	428	2.46	78	-

Reviewers' Comments:

Reviewer #1:

Remarks to the Author:

In the revised manuscript, the authors significantly improved the work by addressing some of the issues raised by the reviewers. However, some important issues should be further resolved. Besides, in light of the new publication by the authors that is related to the current work (but not cited in this work), further analysis is necessary to complement the revised manuscript.

1. Although the authors claimed that the effect of APC on CRTC2 is mostly in the liver due to the higher concentration of APC in the liver compared with other tissues, I would still suggest to check the expression of transcriptional targets of CRTC2/CREB in other tissues. I would suggest the authors to refer to the published works listed in the original comments to find the specific target genes in different tissues other than the liver and adipose tissues.

2. In Figure 5, the authors showed that both Srebf-1a and Srebf-1c expression was reduced by APC. However, these two genes utilize different promoters for the transcription, showing differential regulatory mechanisms. In the liver and adipocytes, Srebf-1c, but not Srebf-1a, is transcriptionally regulated by insulin or feeding, while Srebf-1a is more predominantly expressed in cancer cells. Thus, all the data shown from Figure 5 related to Srebf should be carefully re-written to clarify whether the authors look for the transcriptional regulation of Srebf-1c or Srebf-1a.

3. In Figure 5F, authors utilized primary hepatocytes for the regulation of Srebf-1a and Srebf-2 expression. In normal hepatocytes, the majority of CRTC2 resides in the cytosol and only moves into the nucleus in response to glucagon (or cAMP). How could the authors explain this point?

4. Recently, the authors published an article in cell metabolism (Cell Metabolism. Volume 29, Issue 3, 5 March 2019, Pages 653-667.e6) delineating the role of CREB/CRTC2 in reducing mTORC1 activity under fasting by transcriptional regulation of PER2. As the authors suggested, CRTC2 is active under diet-induced or genetic obesity in the liver irrespective of the feeding status, thus we would speculate that CREB/CRTC2-dependent downregulation of mTORC1 is anticipated under obesity or insulin resistance in the liver. Thus, APC-dependent dissociation of CREB/CRTC2 complex should also affect PER2 expression in the liver, so we would expect to observe higher activity of mTORC1 in the liver. The whole concept is not quite compatible with the current manuscript, and I would assume that is why the authors chose not to cite their own paper in the related work here.

Reviewer #2:

Remarks to the Author:

The manuscript NCOMMS-18-34605 from Prof Liu and colleagues, entitled "A Propolis-Derived Small Molecule Ameliorates Metabolic Syndrome in Obese Mice via Targeting CREB/CRTC2 Transcriptional Complex" reports the discovery that the small molecule artemillin C (APC), from Propolis, disrupts CRTC-CREB-mediated transcription. The authors further show that artemillin C treatment blocks CRTC-CREB-mediated transcription of gluconeogenic genes and LXRalpha. Specifically, the authors find that APC does not impair CREB-CBP association but reduces both a CRTC2-CREB interaction and reducing CBP HAT activity. Treatment of DIO and db/db mice with artemillin C was associated with lowering of body weight, fasting blood glucose, plasma lipids, cholesterol, and liver steatosis. Dose selection is supported with PK and biodistribution studies. The authors link these in vivo effects to artemillin C blockade of CRTC2-CREB transcription of gluconeogenic genes and LXR alpha in liver and hepatocytes. The authors connect their findings with prior published work and find that APC also improves lipid metabolism and increases Fgf21 levels but not in CRTC2 KO mice. Lastly, the authors employ synthetic chemistry to generate a more potent inhibitor A57 that shows similar effects in vivo on gluconeogenic gene expression and fasting blood glucose lowering. These original results and conclusions are well-supported by a broad range of molecular, biochemical, cell-based and in vivo experiments. Additional control experiments were provided to support the author's interpretation of data.

The authors more than sufficiently addressed the review comments I provided. The additional experiments and revisions further enhance the paper mechanistically. The work as a whole is novel and is of broad interest for the metabolic and transcriptional biology research areas and is suitable for publication in Nature Communications.

Reviewer #3:

Remarks to the Author:

In the present manuscript entitled: A Propolis-Derived Small Molecule Ameliorates Metabolic Syndrome in Obese Mice by Targeting the CREB/CRTC2 Transcriptional Complex, the authors focused to investigate the effect of artemillin C, a derivative from the Brazilian green propolis which reduced fasting blood glucose levels in obese mice by disrupting the formation of the CREB/CRTC2 transcriptional complex, and they also used a novel chemical compound, A57, which exhibited higher inhibitory activity on CREB-CRTC2 interaction. The data presented in this manuscript is interesting and important in this field. However, several points should be seriously taken in consideration for the following reasons:

- 1- The authors are advised to revise the manuscript against English language errors that are present throughout the manuscript.
- 2- The authors should respect the use of abbreviations throughout the manuscript.
- 3- The authors measured some parameters such as G6pc and Pck1 in the liver of db/db mice at the mRNA levels, but they should determine the protein levels.
- 4- In the data of figure 3C, the authors used only two concentrations of APC 1 and 5 micro molar, but they should use different concentrations.
- 5- The authors must add n=? in the legends to figures.
- 6- The figures are of very bad resolution and sometimes the protein bands are very smaller than the font of writing.
- 7- I think the authors must revised the statistical analysis for some experiments and a multiple comparison test should be applied.
- 8- Why the authors did not investigate the impact of APC on the signaling of PPAR?
- 9- Did the authors measure the direct effect of APC on cultured beta cells?

Reviewer #4:
Remarks to the Author:

Reviewer #4

In this contribution, Ya-qiong Chen et al. described an artemisinin's novel analog that exhibits higher inhibitory activity on CREB-CRTC2 interaction.

Question 1: The medicinal chemistry part is incomplete which seriously affects the quality of the study.

Response: We are grateful for the reviewer's suggestions in this regard. We have revised the medicinal chemistry sections in our revised manuscript. We added more information on the design of compound **A57**, the SAR analysis of this series of compounds, the racemic and enantiomer of compound **A57**, as well as adding the different inhibitory activities and a pharmacokinetic profiles comparison between **A57** and APC, etc. We hope that this additional information addresses the reviewer's concerns about the medicinal chemistry information.

Reviewer #4 (second revision)

I don't think that adding the *s*-enantiomer and the racemic mixture (as mentioned in my question 4) only made the medicinal chemistry part complete.
The medicinal chemistry part is still incomplete.

Question 2: The design of compound A57 is not explained although the authors indicate that they made a series of molecules from artemisinin as a lead compound!!

Response: To address the reviewer's suggestions, we have added the design of compound **A57** to our revised manuscript. We revised this part as follow: "Initially, we reserved the pharmacophore α,β -unsaturated ketone, and then assessed various substitutions on this pharmacophore in order to increase the inhibitory activity of CREB/CRTC2 protein-protein interactions with APC. A series of novel α,β -unsaturated ketone derivatives were then designed and synthesized (Fig. 8A). Among them, compound **A57** showed better capability for inhibiting CREB/CRTC2 than APC (IC₅₀ 0.74 μ M vs. 24.5 μ M, respectively, Fig. 8B)."

Reviewer #4 (second revision)

The authors indicate that in Figure 8A they show the synthesis of a new series. However, there are no details. Details (nature and positions) of R₂ groups are missing.

I do not believe that with such a general scheme and the paragraph added ("Initially, we reserved the pharmacophore α,β -unsaturated ketone") we can have all the details that led to the discovery of the **A57**.

The arrow used in the scheme (fig 8A) is not the right one, this arrow is used for a retrosynthesis and not for a synthesis.

Question 3: The entire series must be included in the manuscript to fully understand the strategy that led to compound A57. With all series, the effect of substitutions at the 3-position of the phenyl and the effect of the presence of the amide moiety instead of the carboxylic acid can be elucidated

Response: We are grateful to the reviewer's suggestions and comments. We have added the SAR analysis of this series compounds in our revised manuscript. We revised this part in discussion as

follow: “All the synthesized compounds were evaluated *in vitro* for inhibition of CREB/CRTC2. Initially, our SAR started from APC. We then reserved the pharmacophore α,β -unsaturated acid, and assessed the effect of substitutions at the 3-position of the phenyl. The mono-substituted compounds demonstrated a regiochemical preference of para > ortho > meta. Both electron-donating and electron-withdrawing groups on the phenyl ring were tolerated for CREB/CRTC2 inhibition. Among them, phenyl substitution exhibited good inhibitory activity against CREB/CRTC2 interaction. Then, an R1 substitution was explored. Amide compounds displayed better inhibitory activity than carboxylic acid derivatives.”

Reviewer #4 (second revision)

As mentioned in my previous comment, there is no detail regarding the series that has been added!!

The authors indicate that they have made mono, para, ortho and meta substituted molecules with electron-donating and electron-withdrawing groups but with no details on the groups. It remains very general with R2 and R1 !!!

It takes a figure that shows the entire series with all substituents but not only a general text with, R1, R2.

Maybe this series has already been published in a medicinal chemistry journal? Only cite the reference if this is the case.

Question 4: How did the authors arrive with this chiral compound? a racemic mixture will be active or not??

Response: In response to the reviewer’s question, a racemic mixture and another enantiomer were synthesized and we determined their inhibitory activities *in vitro*. The results are described as follows in the revised manuscript (page 20, paragraph 1): “Considering **A57** was a chiral compound, a racemic mixture and another enantiomer were synthesized and we then determined their inhibitory activities *in vitro*. The results revealed that inhibitory activity of **A57** (R-enantiomer) was higher than **A1101** (racemic) or **A58** (S-enantiomer) (Fig. 8C)”.

Reviewer #4 (second revision)

the other enantiomer and the racemic mixture were added. However, it will be necessary to describe their syntheses and their characterizations (NMR, HRMS).

How do the authors explain this difference in activity? Why the R-enantiomer is the most active compared to the S-enantiomer and the racemic mixture?

Question 5: All intermediates must be characterized (¹H, ¹³C, HMRS).

Response: We have added the characterizations (¹H, ¹³C, HMRS) of all intermediates in our revised manuscript (suppl. Fig. 9A).

Reviewer #4 (second revision)

Ok for me.

Question 6: The authors indicate that the goal was to develop “small molecules with satisfactory drug-available”. I find no comparison with artemillin. A57 (ClogP: 6.2) is more lipophilic than artemillin (ClogP: 5.4). This difference in activity is not due to an increase of the lipophilicity?

Response: We are very grateful for the reviewer’s suggestions and comments.

The difference in activity was due to an increase in the lipophilicity. We added a comparison of lipophilicity of compound A57 with APC in our revised manuscript. The pertinent section now reads as follows: “Additionally, A57 (cLog P = 6.2) was more potent than APC (cLog P = 5.4), due to increasing the lipophilicity, and this resulted in higher activity for disrupting CREB/CRTC2 interaction.”

Reviewer #4 (second revision)

The authors claim that lipophilicity explains the difference in activity between APC and compound A57. How to interpret the difference between A57 and the second enantiomer or the racemic mixture which both have the same lipophilicity as A57?

Question 7: A comparison of the pharmacokinetic properties is necessary.

Response: We thank the reviewer for this suggestion. We have added a comparison of the pharmacokinetic properties of compounds A57 and APC. The details are as follow: “We further investigated the bioavailability of A57 and APC administrated orally (*p.o.*) or intravenously (*i.v.*) in ICR mice. The results (Table 1) showed that compound A57 displayed a good AUC (648 ng·h/mL) and better oral bioavailability (25.3% vs 14.4%) compared with APC. This data indicated that the increased effective bioactivity of A57 was tightly associated with increased bioavailability. Collectively, the above data suggested that this series of compounds could serve as possible lead compounds for the development of CREB/CRTC2 inhibitors”.

Reviewer #4 (second revision)

Ok for me.

Responses to Reviewers' Comments

Reviewer #1 (Remarks to the Author):

In the revised manuscript, the authors significantly improved the work by addressing some of the issues raised by the reviewers. However, some important issues should be further resolved. Besides, in light of the new publication by the authors that is related to the current work (but not cited in this work), further analysis is necessary to complement the revised manuscript.

1. Although the authors claimed that the effect of APC on CRTC2 is mostly in the liver due to the higher concentration of APC in the liver compared with other tissues, I would still suggest to check the expression of transcriptional targets of CRTC2/CREB in other tissues. I would suggest the authors to refer to the published works listed in the original comments to find the specific target genes in different tissues other than the liver and adipose tissues.

Response: We are grateful for Reviewer1's suggestions and comments. As suggested by Reviewer1, we checked the transcriptional levels of CREB/CRTC2-specifically regulated genes (e.g., *Nr4a1*, *Pdk1*, *Gcg* and *C2cd4a*, as referred in the papers listed by Reviewer1 in his last comments) in various tissues, including the muscle, brain, spleen, kidney, small intestine and pancreas, of C57 wild type (WT) mice that were fasted for 12 hours, and then orally administered one dose of APC (20 mg/kg) for another 6 hours before being sacrificed. As shown in Suppl. Figure 4i, APC does not significantly affect the mRNA levels of CREB/CRTC2 regulated genes in these tissues, which is consistent with the limited APC distribution in these tissues.

2. In Figure 5, the authors showed that both Srebf-1a and Srebf-1c expression was reduced by APC. However, these two genes utilize different promoters for the transcription, showing differential regulatory mechanisms. In the liver and adipocytes, Srebf-1c, but not Srebf-1a, is transcriptionally regulated by insulin or feeding, while Srebf-1a is more predominantly expressed in cancer cells. Thus, all the data shown from Figure 5 related to Srebf should be carefully re-written to clarify whether the authors look for the transcriptional regulation of Srebf-1c or Srebf-1a.

Response: As suggested by Reviewer1, we re-wrote the sentences about the Srebf-related with Figure 5 to clarify the issue of Srebf-1c or Srebf-1a (page 14, line 11, 18; page 15, line 5, 6 and 11).

3. In Figure 5F, authors utilized primary hepatocytes for the regulation of Srebf-1a and Srebf-2 expression. In normal hepatocytes, the majority of CRTC2 resides in the cytosol and only moves into the nucleus in response to glucagon (or cAMP). How could the authors explain this point?

Response: We agree with Reviewer1's comment that the utilization of primary hepatocytes is not a good model for detecting CREB/CRTC2-mediated transcription of *Srebp1*, as the *in vitro* conditions we used in the experiments (primary hepatocytes treated by insulin) are too simple to mimic *in vivo* circumstance to which hepatocytes are exposed (e.g., exposure to multiple hormones besides insulin, such as GLP-1, dopamine, adrenergic and androgen, which could induce cAMP-PKA-CREB/CRTC2 signaling). Thus, we replaced Figure 5f with *in vivo* data that have shown that APC is incapable of further reducing hepatic mRNA levels of *Srebp1-1c* and *2* in CRTC2 knockout mice, suggesting an essential role of CRTC2 for APC decreasing the transcriptions of *Srebps*.

4. Recently, the authors published an article in cell metabolism (Cell Metabolism. Volume 29, Issue 3, 5 March 2019, Pages 653-667.e6) delineating the role of CREB/CRTC2 in reducing mTORC1 activity under fasting by transcriptional regulation of PER2. As the authors suggested, CRTC2 is active under diet-induced or genetic obesity in the liver irrespective of the feeding status, thus we would speculate that CREB/CRTC2-dependent downregulation of mTORC1 is anticipated under obesity or insulin resistance in the liver. Thus, APC-dependent dissociation of CREB/CRTC2 complex should also affect PER2 expression in the liver, so we would expect to observe higher activity of mTORC1 in the liver. The whole concept is not quite compatible with the current manuscript, and I would assume that is why the authors chose not to cite their own paper in the related work here.

Response: As this manuscript was prepared and submitted much earlier than our work published in *Cell Metabolism* as mentioned by Reviewer1, we have not checked the role of PER2-mTORC1 pathway in APC's functions. In that work, we have reported that CREB/CRTC2 induces the protein amounts of PER2 that recruit TSC1 to inhibit mTORC1 activity. As Reviewer1 suggested, we found that APC indeed reduces CREB-CRTC2S171A induction of the activity of the luciferase reporter driven by the promoter of mouse *Per2* gene (please see below Figure A), which is consistent with our conclusion in the above paper, in which CREB/CRTC2 has been shown to increase *Per2* expression.

We then checked the effects of APC on mTORC1 activation as indicated by the phosphorylation levels of S6K, one of its target proteins, in primary hepatocytes pretreated by glucagon for 4 hours and followed by 12-hour insulin incubation. As shown in figure B below, APC enhances mTORC1 activity as indicated by the increase of S235/236 phosphorylation level of S6K, whereas, APC still reduces the protein amounts of mature

SREBP1 (m-SREBP1) and fat acid synthase (FASN) in glucagon-insulin treated cells, which suggests that APC is capable of reducing lipogenic gene expression when it increases mTORC1 activity at the same time. mTORC1 has been reported to enhance SREBP1c mediated lipogenesis mainly by increasing the maturation of SREBP1c proteins (2015, *The CREB coactivator CRTC2 controls hepatic lipid metabolism by regulating SREBP1*, *Nature*, Aug 13;524(7564):243-6.). However, a related work has demonstrated that mTORC1 hyperactivation caused by liver specific TSC1 deletion is insufficient to induce SREBP1c activation and de novo lipogenesis in the liver of high fat diet fed mice (*Akt stimulates hepatic SREBP1c and lipogenesis through parallel mTORC1-dependent and independent pathways*. *Cell Metabolism*. 2011. 14.1: 21-32). Considered the requirement of TSC1 for PER2 to inhibit mTORC1 activity, the reduction of PER2 by APC may not be sufficient to induce SREBP1c activation. Moreover, our data suggest that APC reduces the transcription of SREBP1c in vivo (Figures 5b and 5g), which would result in the decrease of whole SREBP1c protein levels and overcome the effect on lipogenesis caused by APC activation of mTORC1. Taken together, we think that APC activation of mTORC1 by reducing PER2 expression is compatible with its suppressing effects on hepatic lipogenesis.

A

Figure A. APC reduces CREB/CRTC2 induced *Per2*-Luc activity. HEK293 cells were transfected with a plasmid of a luciferase reporter driven by mouse *Per2* promoter (pGL3-m*Per2*-Luciferase) and RSV- β -Gal, together with either a control plasmid or overexpression plasmids of FLAG-CREB and HA-CRTC2-S171A. Six hours after transfection, the cells were treated with APC (1, 5 or 25 μ M) for another 14 hours before collection. Luciferases reporter activities were normalized by β -Gal signals. Data are represented as mean \pm SEM (n=4). *, $p < 0.05$; **, $p < 0.01$. p values were determined by one-way ANOVA followed Dunnett's multiple comparisons test (Left). The over-expressed proteins FLAG-CREB and HA-CRTC2-171A were detected in the lysates of the luciferase reporter assay by immunoblotting (Right).

B

Figure B. Immunoblotting analysis of phosphorylated or total protein amounts of S6K, mature SREBP-1 and FASN in primary hepatocytes. The cells were pretreated with APC (10 μ M) for 2 hours, exposed to glucagon (100 nM) for 4 hours, and then followed by 12-hour insulin (100 nM) incubation before collection.

Reviewer #2 (Remarks to the Author):

The manuscript NCOMMS-18-34605 from Prof Liu and colleagues, entitled "A Propolis-Derived Small Molecule Ameliorates Metabolic Syndrome in Obese Mice via Targeting CREB/CRTC2 Transcriptional Complex" reports the discovery that the small molecule artemillin C (APC), from Propolis, disrupts CRTC-CREB-mediated transcription. The authors further show that artemillin C treatment blocks CRTC-CREB-mediated transcription of gluconeogenic genes and LXRalpha. Specifically, the authors find that APC does not impair CREB-CBP association but reduces both a CRTC2-CREB interaction and reducing CBP HAT activity. Treatment of DIO and db/db mice with artemillin C was associated with lowering of body weight, fasting blood glucose, plasma lipids, cholesterol, and liver steatosis. Dose selection is supported with PK and biodistribution studies. The authors link these in vivo effects to artemillin C blockade of CRTC2-CREB transcription of gluconeogenic genes and LXR alpha in liver and hepatocytes.

The authors connect their findings with prior published work and find that APC also improves lipid metabolism and increases Fgf21 levels but not in CRTC2 KO mice. Lastly, the authors employ synthetic chemistry to generate a more potent inhibitor A57 that shows similar effects in vivo on gluconeogenic gene expression and fasting blood glucose lowering. These original results and conclusions are well-supported by a broad range of molecular, biochemical, cell-based and in vivo experiments. Additional control experiments were provided to support the author's interpretation of data.

The authors more than sufficiently addressed the review comments I provided. The additional experiments and revisions further enhance the paper mechanistically. The work as a whole is novel and is of broad interest for the metabolic and transcriptional biology research areas and is suitable for publication in Nature Communications.

Response: We are very grateful for Review2's comments. In particular, we would like to sincerely appreciate the time and effort of Reviewer2 to help us improve the quality of this manuscript.

Reviewer #3 (Remarks to the Author):

In the present manuscript entitled: A Propolis-Derived Small Molecule Ameliorates Metabolic Syndrome in Obese Mice by Targeting the CREB/CRTC2 Transcriptional Complex, the authors focused to investigate the effect of artemillin C, a derivative from the Brazilian green propolis which reduced fasting blood glucose levels in obese mice by disrupting the formation of the CREB/CRTC2 transcriptional complex, and they also used a novel chemical compound, A57, which exhibited higher inhibitory activity on CREB-CRTC2 interaction. The data presented in this manuscript is interesting and important in this field. However, several points should be seriously taken in consideration for the following reasons:

- 1- The authors are advised to revise the manuscript against English language errors that are present throughout the manuscript.

Response: We are grateful for Reviewer3's comments. Our manuscript has been revised by a native English editor. We attached the certificate received from that native English editor.

Certificate of English Language Editing

Manuscript Title:
A Propolis-Derived Small Molecule Ameliorates Metabolic Syndrome in Obese Mice by Targeting the CREB/CRTC2 Transcriptional Complex

Date of Revision:
January 14, 2021

Abstract:

Although the beneficial effects of the natural medicine propolis on metabolic syndrome are well-known, its molecular targets and the underlying mechanisms mediating this are not fully understood. Here, we report that Brazilian green propolis reduces fasting blood glucose levels in obese mice by disrupting the formation of the CREB/CRTC2 transcriptional complex, a key regulator of hepatic gluconeogenesis. By using a mammalian two-hybrid system to detect the association between CREB and CRTC2, we screened isolated chemical compounds from Brazilian green propolis to identify a small molecule, APC that is capable of blocking...

This document certifies that the manuscript listed above was copy edited for proper English language at LetPub. All of our language editors are native English speakers with long-term experience in editing scientific and technical manuscripts. We are committed to leveling the playing field for researchers whose native language is not English.

- Neither the research content nor the authors' intended meaning were altered in any way during the editing process.
- Documents receiving this certification should be considered ready for publication where language issues are concerned. However, the authors may accept or reject LetPub's suggestions and changes at their own discretion.
- If you have any questions or concerns about this edited document, please contact us at support@letpub.com

LetPub is an author service brand owned and operated by Accdon LLC. Headquartered in the Boston area, we are a full-spectrum author services company with a large team of US-based certified language and scientific editors, ISO 17001 accredited translators, and professional scientific illustrators and animators. We advocate ethical publication practices and are an official member of the Committee on Publication Ethics (COPE).

For more information about our company, services, and partnership programs, please visit www.letpub.com.
© 2020 Accdon, LLC. All Rights Reserved. Tel: 1-781-202-9988 Email: info@accdon.com Address: 400 Fifth Ave, Suite 530, Waltham, MA 02451, United States

2- The authors should respect the use of abbreviations throughout the manuscript.

Response: As suggested by Reviewer3, we have added abbreviation of G6p-luc, DIO and OGTT in the section of abbreviation (page 3, line 13, 17 and 18). Moreover, we have revised the non-standard abbreviations in the manuscript.

3- The authors measured some parameters such as G6pc and Pck1 in the liver of db/db mice at the mRNA levels, but they should determine the protein levels.

Response: As suggested by Reviewer3, we checked the protein levels of PCK1 and PGC1 α in the liver of *db/db* mice and found that APC reduces both levels of these two proteins (Supp. Figure 7B). Additionally, we also tried several anti-G6PC antibodies, whereas, we did not find a reliable one to detect G6PC proteins in mouse liver. We think that the above results are sufficient to support our conclusion that APC decreases the expression of glucogenic proteins in mouse liver.

4- In the data of figure 3C, the authors used only two concentrations of APC 1 and 5 micro molar, but they should use different concentrations.

Response: As suggested by Reviewer3, we checked the glucose output from primary hepatocytes with more concentrations of APC (e.g. 1, 5, 30 μ M) and got the consistent data that were shown in Figure 3c in revised manuscript.

5- The authors must add n=? in the legends to figures.

Response: As suggested by Reviewer3, we added corresponding n values in all the figure legends, for example the legends of Figures, “n=5-7 mice per group” in Figure 1a (page 43 line 6), and “n=3 per treatment” in Figure 3e (page 51 line 4).

6- The figures are of very bad resolution and sometimes the protein bands are very smaller than the font of writing.

Response: As suggested by Reviewer3, we improved the quality and resolution of figures. We also enlarged the size of western blotting pictures in the revised manuscript, for example the pictures in Figures 1d and Figure 2d, 2g, 2h.

7- I think the authors must revised the statistical analysis for some experiments and a multiple comparison test should be applied.

Response: As suggested by Reviewer3, we re-performed statistical analyses for all the experiments. As indicated in the corresponding legends, the data were statistically analyzed by using either unpaired two-tailed Student's *t*-test with Welch's correction between two groups (e.g. Figure 1b right and Figure 7d), or one-way ANOVA followed Dunnett's

multiple comparisons test among groups more than three compared with single control (e.g. Figure 1c and Figure 3b), or two-way ANOVA followed Bonferroni's multiple comparisons test among different groups (e.g. Figure 1d right and Figure 3f). Graphpad Prism software (8.0) was used for data statistical analysis.

8- Why the authors did not investigate the impact of APC on the signaling of PPAR?

Response: We are grateful for Reviewer3's comments. Actually, we have explored possible involvement of PPARs in APC's effects. Our data have shown that APC does not alter the induced activity of PPRE-luc (PPAR response element driven luciferase reporter) by the overexpression of PPAR α , RXR α or LXR α (Suppl. Figure 6c), which suggests that APC is not an agonist or antagonist of these nuclear receptors. Moreover, APC does not change the mRNA levels of PPAR α and PPAR γ in vivo (Figure 6a). Taken together, we think that the impact of APC on PPAR pathway is not significant.

9- Did the authors measure the direct effect of APC on cultured beta cells?

Response: We are grateful for Reviewer3's comments. Since our in vivo data have shown that the distribution of APC in mouse pancreas is very low (Figure 4k) and APC does not significantly change the transcription of CREB/CRTC2 regulated genes in mouse pancreas (supply Figure. 4i), we think that the direct effect of APC on beta cells would be minimum. Therefore, we do not check APC's effect on the cultured beta cells.

Reviewer #4 (Remarks to the Author):

In this contribution, Ya-qiong Chen et al. described an artemisinin's novel analog that exhibits higher inhibitory activity on CREB-CRTC2 interaction.

Question 1: The medicinal chemistry part is incomplete which seriously affects the quality of the study.

Response: We are grateful for Reviewer4's comments. As suggested by Reviewer4, we have revised the medicinal chemistry sections in our revised manuscript. We added more information on the design of compound **A57**, the SAR analysis of this series of compounds, the racemic and enantiomer of compound **A57**, as well as adding the different inhibitory activities and a pharmacokinetic profiles comparison between **A57** and APC, etc. We hope that this additional information addresses the reviewer's concerns about the medicinal chemistry information.

Reviewer #4 (second revision)

I don't think that adding the s-enantiomer and the racemic mixture (as mentioned in my question 4) only made the medicinal chemistry part complete. The medicinal chemistry part is still incomplete.

Response: We are grateful for Reviewer4's comments. As suggested by Reviewer4, we have added the more data in the medicinal chemistry part in our revised manuscript and revised supporting information. The medicinal chemistry part was revised as follow:

Developing novel CREB/CRTC2 inhibitors using APC as a lead compound

As artemisinin C (APC) displayed remarkable bioactivity to block CREB-CRTC2 protein-protein interaction and enhance insulin sensitivity *in vivo* (Fig. 1-7 and suppl. Fig. 1-7), we decided to design and discovery more potent inhibitor of CREB/CRTC2 (Fig. 8 and suppl. Fig. 9-10 and suppl. Table S4). Initially, the inhibitory activity of series in-house compounds with similar structure were detected by our CREB-CRTC2 two hybrid system. Fortunately, compound **A32** was discovered with an IC₅₀ value of 9.95 μM, which displayed better inhibitory activity than APC. Then, based on the structure of APC and compound **A32**, a series of novel compounds were designed and synthesized in order to increase the inhibitory activity of CREB/CRTC2 protein-protein interaction. (Fig. 8a, suppl. Fig. 9, suppl. Fig. 10a, and suppl. Table S4). All the synthesized compounds were evaluated *in vitro* for inhibition of CREB/CRTC2. Firstly, we reserved the 2-CF₃ group as R², R¹ substitution was explored. Amide compounds displayed better inhibitory activity than carboxylic acid and ester derivatives. Then, we reserved pharmacophore α,β-unsaturated amide, and assessed the effect of substitutions of the phenyl. Both the electron-donating and electron-withdrawing groups on the phenyl ring were tolerated for CREB/CRTC2 inhibition. The mono-substituted compounds demonstrated a regiochemical preference of ortho ≈ meta > para. When the R² group is 3-OMe, compound **A54** was afforded, the inhibitory activity of CREB/CRTC2 protein-protein interaction increased with an IC₅₀ value of 1.5 μM, which is 16-fold more potent than APC. Finally, when R¹ group is replaced by (S)-

4-phenyloxazolidin-2-one and R² group is 3-Ph, compound **A57** was obtained, which showed better capacity for inhibiting CREB/CRTC2 than APC *in vitro* (IC₅₀ 0.74 μM vs. 24.5 μM, respectively, Fig. 8b). Considering **A57** was a chiral compound, a racemic mixture **A1101** and another enantiomer **A58** were synthesized and their inhibitory activity *in vitro* was determined. Our results revealed that the inhibitory activity of **A57** (*S*-enantiomer) was higher than **A1101** (racemic) and **A58** (*R*-enantiomer) (Fig. 8c). These results suggest that the configuration of the phenyl group is related to the inhibitory activity. Additionally, **A57** (cLog P = 6.2) was more potent than APC (cLog P = 5.4), due to the increased lipophilicity, and resulted in higher activity disrupting the CREB/CRTC2 interaction. Furthermore, treatment with 50 μM **A57** caused little damage to HEK293T cells (Suppl. Fig. 10b), excluding cellular toxicity from the inhibitory activity of **A57**. In common with APC, **A57** had little effect on CREB phosphorylation in primary hepatocytes (Suppl. Fig. 10c).

Table S4. The structure of CREB/CRTC2 inhibitors and their inhibitory activity.

Compd.	R ¹	R ²	IC ₅₀ (μM)
A32		2-CF₃	9.95
A35		2-CF₃	53.3
A37		2-CF₃	> 100
A40	-OH	2-CF₃	> 100
A43	-OMe	2-CF₃	81.1
A47		3-CF₃	13.8
A50		4-CF₃	18.7
A53		2-OMe	3.1
A54		3-OMe	1.5
A56		4-OMe	39.7
A57		3-Ph	0.74
A58		3-Ph	73.5
A1101		3-Ph	3.56

(supplementary material page 45)

Question 2: The design of compound A57 is not explained although the authors indicate that they made a series of molecules from artepillin as a lead compound!!

Response: We are grateful for Reviewer4's comments. To address Reviewer4's comments, we have added the design of compound A57 to our revised manuscript. We revised this part as follow: "Initially, we reserved the pharmacophore α,β -unsaturated ketone, and then assessed various substitutions on this pharmacophore in order to increase the inhibitory activity of CREB/CRTC2 protein-protein interactions with APC. A series of novel α,β -unsaturated ketone derivatives were then designed and synthesized (Figure 8a). Among them, compound A57 showed better capability for inhibiting CREB/CRTC2 than APC (IC₅₀:0.74 μ M vs. 24.5 μ M, respectively, Figure 8b)."

Reviewer #4 (second revision)

The authors indicate that in Figure 8A they show the synthesis of a new series. However, there are no details. Details (nature and positions) of R2 groups are missing. I do not believe that with such a general scheme and the paragraph added ("Initially, we reserved the pharmacophore α,β -unsaturated ketone") we can have all the details that led to the discovery of the A57.

The arrow used in the scheme (Fig. 8a) is not the right one, this arrow is used for a retrosynthesis and not for a synthesis.

Response: We are grateful for Reviewer4's comments. As suggested by Reviewer4, we have added detailed information of this novel series compounds in the revised manuscript and supporting information (supplementary material page 23-35). We have added a Table 4 (supplementary material page 45) in our revised supporting information to illustrate the structure of these novel CREB/CRTC2 interaction inhibitors and inhibitory activity of these compounds.

The arrow used in the Scheme (Fig 8a) has been revised as follow:

Question 3: The entire series must be included in the manuscript to fully understand the strategy that led to compound A57. With all series, the effect of substitutions at the 3-position of the phenyl and the effect of the presence of the amide moiety instead of the carboxylic acid can be elucidated.

Response: We are grateful for Reviewer4's comments. As suggested by Reviewer4, we have added the SAR analysis of this series compounds in our revised manuscript. We have revised this part in discussion as follow: "All the synthesized compounds were evaluated in vitro for inhibition of CREB/CRTC2. Initially, our SAR started from APC. We then reserved the pharmacophore α,β -unsaturated acid, and assessed the effect of substitutions at the 3-position of the phenyl. The mono-substituted compounds demonstrated a regiochemical preference of para > ortho > meta. Both electron-donating-and electron-withdrawing groups on the phenyl ring were tolerated for CREB/CRTC2inhibition. Among them, phenyl substitution exhibited good inhibitory activity against CREB/CRTC2 interaction. Then, an R1 substitution was explored. Amide compounds displayed better inhibitory activity than carboxylic acid derivatives." (page 28 line16 to page 29 line 10)

Reviewer #4 (second revision)

As mentioned in my previous comment, there is no detail regarding the series that has been added!!

The authors indicate that they have made mono, para, ortho and meta substituted molecules with electron-donating and electron-withdrawing groups but with no details on the groups. It remains very general with R2 and R1 !!!

It takes a figure that shows the entire series with all substituents but not only a general text with, R1, R2.

Maybe this series has already been published in a medicinal chemistry journal? Only cite the reference if this is the case.

Response: We are grateful for Reviewer4's comments. This series of compounds have not been published in any journals. As suggested by Reviewer4, we have added detailed information of this novel series compounds in the revised manuscript and supporting information (supplementary material page 45, Table 4). We have added a supplementary table 4 in our revised supporting information to illustrate the structures of these novel CREB/CRTC2 interaction inhibitors and inhibitory activity of these compounds.

Question 4: How did the authors arrive with this chiral compound? a racemic mixture will be active or not??

Response: We are grateful for Reviewer4's comments. In response to this Reviewer4's question, we synthesized a racemic mixture and another enantiomer, and then determined their inhibitory activities in vitro. We found that the inhibitory activity of A57 (R-enantiomer) is higher than that of A1101 (racemic) or A58 (S-enantiomer) (Figure 8C).

Reviewer #4 (second revision)

The other enantiomer and the racemic mixture were added. However, it will be necessary to describe their syntheses and their characterizations (NMR, HRMS).

How do the authors explain this difference in activity? Why the *S*-enantiomer is the most active compared to the *R*-enantiomer and the racemic mixture?

Response: We are grateful to the reviewer's suggestions and comments.

(1) We have added syntheses and their characterizations of another enantiomer **A58** and racemic compound **A1101** (^1H NMR, ^{13}C NMR, and HRMS) in our revised supporting information (supplementary material page 29-34).

(2) We added the related information in the revised manuscript as follow (supplementary material page 34-35):

The *S*-configured enantiomer **A57** ($\text{IC}_{50} = 0.74 \mu\text{M}$) was more effective than the *R*-configured compound **A58** ($\text{IC}_{50} = 73.5 \mu\text{M}$) and the racemate **A1101** ($\text{IC}_{50} = 3.56 \mu\text{M}$). These results suggest that the configuration of the phenyl group is related to the inhibitory activity. The configuration of compounds **A57** and **A58** was shown in suppl. Fig. 9e. The different configuration of the phenyl group in these two compounds explains why compound **A57** has better inhibitory activity than **A58**. For the *S*-configured compound **A57**, phenyl group in compound **A57** may occupy the active pocket of protein CREB, which increase the inhibitory activity. Consistent with the poorer inhibitory activity to CREB/CRTC2 interaction, compound **A58** exhibits a different configuration. It is possible that the steric bulk of the phenyl group in compound **A58** hinder interaction between the compound **A58** and the protein CREB, which reduced the inhibitory activity.

Supply Figure 9e. The configuration of compounds **A57** (A) and **A58** (B).

Question 5: All intermediates must be characterized (^1H , ^{13}C , HMRS).

Response: We have added the characterizations (^1H , ^{13}C , HMRS) of all intermediates in our revised manuscript (suppl. Fig. 9a).

Reviewer #4 (second revision)

Ok for me.

Question 6: The authors indicate that the goal was to develop “small molecules with satisfactory drug-available”. I find no comparison with artepillin. **A57 (ClogP: 6.2) is more lipophilic than artepillin (ClogP: 5.4). This difference in activity is not due to an**

increase of the lipophilicity?

Response: We are grateful for Reviewer4's comments. We believe that the difference in activity is due to an increase in the lipophilicity. We added a comparison of lipophilicity of compound **A57** with APC in our revised manuscript. The pertinent section now reads as follows (page 21, line 9-11): "Additionally, **A57** (cLog P = 6.2) was more potent than APC (cLog P = 5.4), due to increasing the lipophilicity, and this resulted in higher activity for disrupting CREB/CRTC2 interaction."

Reviewer #4 (second revision)

The authors claim that lipophilicity explains the difference in activity between APC and compound **A57**. How to interpret the difference between **A57** and the second enantiomer or the racemic mixture which both have the same lipophilicity as **A57**?

Response: We are grateful to the reviewer's suggestions and comments. Compounds **A57**, **A58** and **A1101** have the same lipophilicity, however, the configuration of these three compounds is different. The configuration of compounds **A57** and **A58** was shown in suppl. Figure 9e. The different configuration of the phenyl group in these two compounds explains why compound **A57** has better inhibitory activity than **A58**. For the *S*-configured compound **A57**, phenyl group in compound **A57** may occupy the active domain of protein CREB, which increase the inhibitory activity. Consistent with the poorer inhibitory activity to CREB/CRTC2 interaction, compound **A58** exhibits a different configuration. It is possible that the steric bulk of the phenyl group in compound **A58** hinder interaction between the compound **A58** and the protein CREB, which reduced the inhibitory activity.

Figure 9e. The configuration of compounds **A57** (A) and **A58** (B).

Question 7: A comparison of the pharmacokinetic properties is necessary.

Response: We are grateful for Reviewer4's comments. We have added a comparison of the pharmacokinetic properties of compounds **A57** and APC. The details are as follow: "We further investigated the bioavailability of **A57** and APC administrated orally (p.o.) or intravenously (i.v.) in ICR mice. The results (Table 1) showed that compound **A57** displayed a good AUC (648 ng·h/mL) and better oral bioavailability (25.3% vs 14.4%) compared with APC. This data indicated that the increased effective bioactivity of **A57** was tightly associated with increased bioavailability. Collectively, the above data suggested that this series of compounds could serve as possible lead compounds for the development of CREB/CRTC2 inhibitors".

Reviewer #4 (second revision)

Ok for me.

Reviewers' Comments:

Reviewer #1:

Remarks to the Author:

I have a few more points that have to be addressed in the revised work.

1. In Figure B, it appeared that APC-induced p-S6K level was rather inhibited by glu/ins treatment (lane 3 vs lane 4). Is it significant?
2. The authors showed that APC-dependent activation of mTORC1 activity is independent of its effect on reducing SREBP-1c expression and the subsequent repression of lipogenesis. The authors should simply run western blot for mTORC1 signaling by utilizing the existing samples.
3. Plus, it is desirable to cite and incorporate the recent work by the authors (Cell Metabolism. Volume 29, Issue 3, 5 March 2019, Pages 653-667.e6) at least in the discussion. Perhaps the authors could use the argument written in the response.

Reviewer #4:

Remarks to the Author:

The authors have answered all of my comments and questions.

REVIEWER COMMENTS

Reviewer #1 (Remarks to the Author):

I have a few more points that have to be addressed in the revised work.

1. In Figure B, it appeared that APC-induced p-S6K level was rather inhibited by glu/ins treatment (lane 3 vs lane 4). Is it significant?

Response: We agree with Reviewer1's comments that the western blotting band of p-S6K level of lane 4 seems weaker than that of lane3 in Figure B. However, we performed the densitometry analysis of all the results from three independent experiments and found that the difference between lane3 and 4 was mild and not significant (Figure 1B).

2. The authors showed that APC-dependent activation of mTORC1 activity is independent of its effect on reducing SREBP-1c expression and the subsequent repression of lipogenesis. The authors should simply run western blot for mTORC1 signaling by utilizing the existing samples.

Response: We are grateful for Reviewer1's comments. As Reviewer1 suggested, we had run western blot (Figure 1A) and performed corresponding densitometry analysis (Figure 1B) for mTORC1 signaling (including p-S6K, p-S6 and p-4E-BP1).

Figure 1. Immunoblotting analysis mTORC1 signaling in primary hepatocytes. The cells were pretreated with APC (10 μ M) for 2 hours, exposed

to glucagon (100 nM) for 4 hours, and then followed by 12-hour insulin (100 nM) incubation before collection. The relative phosphorylation level of mTOR, S6K, S6 and 4E-BP1 were normalized by total protein levels respectively. The relative protein levels of SREBP-1c and FASN were normalized by ACTION protein levels. One of three independent experiments was shown here. Data are represented as mean \pm SEM. *, $p < 0.05$; **, $p < 0.01$; p values were determined by two-way ANOVA followed Bonferroni's multiple comparisons test.

3. Plus, it is desirable to cite and incorporate the recent work by the authors (Cell Metabolism. Volume 29, Issue 3, 5 March 2019, Pages 653-667.e6) at least in the discussion. Perhaps the authors could use the argument written in the response.

Response: We are grateful for Reviewer1's comments. In the current manuscript, we have cited, incorporated and discussed this recent work in the Discussion (please see page 28 line 13).

Reviewer #4 (Remarks to the Author):

The authors have answered all of my comments and questions.

Reviewers' Comments:

Reviewer #1:

Remarks to the Author:

The authors sufficiently provided additional data and comments that were suggested the reviewer.

RESPONSE TO REVIEWERS' COMMENTS

Reviewer #1 (Remarks to the Author):

The authors sufficiently provided additional data and comments that were suggested by reviewer.

Response: We are grateful for Review 1's comments. Moreover, we would like to sincerely appreciate the time and efforts to help us improve the quality of this manuscript.